# DiffusionSat: A Generative Foundation Model for Satellite Imagery

**Samar Khanna**[1*], **Patrick Liu**[1], **Linqi Zhou**[1], **Chenlin Meng**[1], **Robin Rombach**[2], **Marshall Burke**[1], **David B. Lobell**[1], **Stefano Ermon**[1,3]

[1]Stanford University, [2]Stability AI, [3]CZ Biohub
[*]Correspondence to: `samarkhanna [at] cs.stanford.edu`

## Abstract

Diffusion models have achieved state-of-the-art results on many modalities including images, speech, and video. However, existing models are not tailored to support remote sensing data, which is widely used in important applications including environmental monitoring and crop-yield prediction. Satellite images are significantly different from natural images – they can be multi-spectral, irregularly sampled across time – and existing diffusion models trained on images from the Web do not support them. Furthermore, remote sensing data is inherently spatio-temporal, requiring conditional generation tasks not supported by traditional methods based on captions or images. In this paper, we present DiffusionSat, to date the largest generative foundation model trained on a collection of publicly available large, high-resolution remote sensing datasets. As text-based captions are sparsely available for satellite images, we incorporate the associated metadata such as geolocation as conditioning information. Our method produces realistic samples and can be used to solve multiple generative tasks including temporal generation, superresolution given multi-spectral inputs and in-painting. Our method outperforms previous state-of-the-art methods for satellite image generation and is the first large-scale *generative* foundation model for satellite imagery. The project website can be found here: `https://samar-khanna.github.io/DiffusionSat/`

## 1 Introduction

Diffusion models have achieved state of the art results in image generation (Sohl-Dickstein et al., 2015; Ho et al., 2020; Dhariwal & Nichol, 2021; Kingma et al., 2021; Song & Ermon, 2019; 2020). Large scale models such as Stable Diffusion Rombach et al. (2022) (SD) have been trained on Internet-scale image-text datasets to generate high-resolution images from user-provided captions. These diffusion-based foundation models, used as priors, have led to major improvements in a variety of inverse problems like inpainting, colorization, deblurring (Luo et al., 2023), medical image reconstruction (Khader et al., 2023; Xie & Li, 2022), and video generation (Blattmann et al., 2023).

Similarly, there are a variety of high-impact ML tasks involving the analysis of satellite images, such as disaster response, environmental monitoring, poverty prediction, crop-yield estimation, urban planning and others (Gupta et al., 2019; Burke et al., 2021; Ayush et al., 2021b; 2020; Jean et al., 2016; You et al., 2017; Wang et al., 2018; Rußwurm & Körner, 2020; Martinez et al., 2021; M Rustowicz et al., 2019; Yeh et al., 2021). These tasks consist of important inverse problems, such as super-resolution (from frequent low resolution images to high resolution ones), cloud removal, temporal in-painting and more. However, satellite images fundamentally differ from natural images in terms of perspective, resolutions, additional spectral bands, and temporal regularity. While foundation models have been recently developed for discriminative learning on satellite images Cong et al. (2022); Ayush et al. (2021a); Bastani et al. (2022), they are not designed to and cannot solve the inverse problems (eg: super-resolution) described above.

To fill this gap, we propose **DiffusionSat**, a generative foundation model for satellite imagery inspired from SD. Using commonly associated metadata with satellite images including latitude, longitude, timestamp, and ground-sampling distance (GSD), we train our model for single-image generation on a collection of publicly available satellite image data sets. Further, inspired from

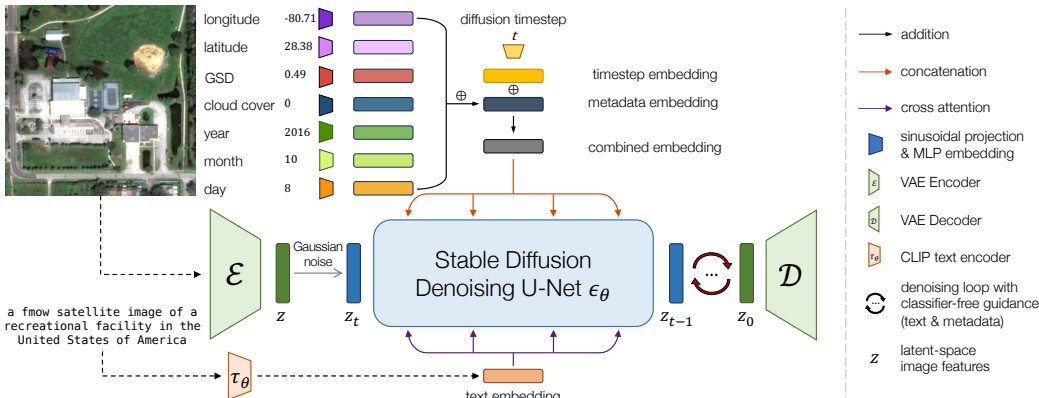

Figure 1: Conditioning on freely available metadata and using large, publicly available satellite imagery datasets shows DiffusionSat is a powerful generative foundation model for remote sensing data.

ControlNets Zhang & Agrawala (2023), we design conditioning models that can easily be trained for specific generative tasks or inverse problems including super-resolution, in-painting, and temporal generation. Specifically, our contributions include:

1. We propose a novel generative foundation model for satellite image data with the ability to generate high-resolution satellite imagery from numerical metadata as well as text.

2. We design a novel 3D-conditioning extension which enables DiffusionSat to demonstrate state-of-the-art performance on super-resolution, temporal generation, and in-painting

3. We collect and compile a global generative pre-training dataset from large, publicly available satelilte image datasets (see section 3.1).

## 2 BACKGROUND

**Diffusion Models** Diffusion models are generative models that aim to learn a data distribution $p_{\text{Data}}$ from samples (Sohl-Dickstein et al., 2015; Ho et al., 2020; Song & Ermon, 2019; Song et al., 2020b; Song & Ermon, 2020). Given an input image $x \sim p_{\text{Data}}$, we add noise to create a *noisy* input $x_t = \alpha_t x + \sigma_t \epsilon$, where $\epsilon \sim \mathcal{N}(\mathbf{0}, \mathbf{I})$ is Gaussian noise. $\alpha_t$ and $\sigma_t$ denote a noise schedule parameterized by diffusion time $t$ (higher $t$ leads to more added noise). The diffusion model $\epsilon_\theta$ then aims to *denoise* $x_t$, and is optimized using the score-matching objective:

$$\mathbb{E}_{x \sim p_{\text{Data}}, \epsilon \sim \mathcal{N}(\mathbf{0}, \mathbf{I})} \left[ ||y - \epsilon_\theta(x_t; t, c)||_2^2 \right] \tag{1}$$

where the target $y$ can be the input noise $\epsilon$, the input image $x$ or the "velocity" $v = \alpha_t \epsilon - \sigma_t x$. We can additionally condition the denoising model with side information $\boldsymbol{c} \in \mathbb{R}^D$, which can be a class embedding, text, or other images etc.

Latent diffusion models (LDMs) (Vahdat et al., 2021; Sinha et al., 2021; Rombach et al., 2022) first downsample the input $x$ using a VAE with an encoder $\mathcal{E}$ and a decoder $\mathcal{D}$, such that $\tilde{x} = \mathcal{D}(\mathcal{E}(x))$ is a reconstructed image. Instead of denoising the input image $x$, the diffusion process is used on a downsampled latent representation $z = \mathcal{E}(x)$ This approach reduces computational and memory cost and has formed the basis for the popularly used StableDiffusion (SD) model (Rombach et al., 2022).

## 3 METHOD

First, we describe our method for the following tasks of interest: single-image generation, conditioned on text and metadata, multi-spectral superresolution, temporal prediction, and temporal inpainting.

### 3.1 SINGLE IMAGE GENERATION

Our first goal is to pre-train DiffusionSat to be able to generate *single* images given an input text prompt and/or metadata. Concretely, we begin by considering datasets such that each image

$\mathbf{x} \in \mathbb{R}^{C \times H \times W}$ is paired with an associated text-caption $\tau$. Our goal is to learn the conditional data distribution $p(\mathbf{x}|\tau)$ such that we can sample new images $\tilde{\mathbf{x}} \sim p(\cdot|\tau)$.

LDMs are popularly used for text-to-image generation primarily for their strong ability to use text prompts. An associated text prompt $\tau$ is tokenized, encoded via CLIP (Radford et al., 2021), and then passed to the DM $\epsilon_\theta(\mathbf{x}_t; t, \tau)$ via cross-attention (Vaswani et al., 2017) at each layer. However, while text prompts are widely available for image datasets such as LAION-5B (Schuhmann et al., 2022), satellite images typically either do not have such captions, or are accompanied by object-detection boxes, segmentation masks, or classification labels. Moreover, requiring such labels precludes the use of vast amounts of unlabeled satellite imagery. Ideally, we would like to pretrain DiffusionSat on existing labelled and unlabelled datasets, without necessarily curating expensive labels.

To solve this challenge, we note that satellite images are commonly associated with metadata including their timestamp, latitude, longitude, and various other numerical information that are correlated with the image (Christie et al., 2018). We thus consider datasets where each image $\mathbf{x} \in \mathbb{R}^{C \times H \times W}$ is paired with a text-caption $\tau$, as well as cheaply available numerical metadata $\mathbf{k} \in \mathbb{R}^M$, where $M$ is the number of metadata items. We thus wish to learn the data distribution $p(\mathbf{x}|\tau, \mathbf{k})$. With good enough metadata $\mathbf{k}$, we want to still sample an image of high quality even if $\tau$ is poor or missing.

We now turn to conditioning on $\mathbf{k}$. One option is to naively incorporate each numerical metadata item $k_j$, $j \in \{1, \ldots, M\}$, into the text caption with a short description. However, this approach unnecessarily discretizes continuous-valued covariates and can suffer from text-encoders' known shortcomings related to encoding numerical information (Radford et al., 2021). Instead, we choose to encode the metadata using the same sinusoidal timestep embedding eq. (2) used in diffusion models:

$$\texttt{Project}(k, 2i) = \sin\left(k\Omega^{-\frac{2i}{d}}\right), \ \texttt{Project}(k, 2i+1) = \cos\left(k\Omega^{-\frac{2i}{d}}\right) \tag{2}$$

where $k$ is the metadata or timestep value, $i$ is the index of feature dimension in the encoding, $d$ is the dimension, and $\Omega = 10000$ is a large constant. Each metadata value $k_j$ is first normalized to a value between 0 and 1000 (since the diffusion timestep $t \in \{0, \ldots, 1000\}$), and is then projected via the sinusoidal encoding. A different MLP for each metadatum encodes the projected metadata value identically to the diffusion timestep $t$ (Ho et al., 2020) as follows eq. (3):

$$f_{\theta_j}(k_j) = \texttt{MLP}\left([\texttt{Project}(k_j, 0), \ldots, \texttt{Project}(k_j, d)]\right) \tag{3}$$

where $f_{\theta_j}$ represents the learned MLP embedding for metadata value $k_j$, corresponding to metadata type $j$ (eg: longitude). Our embedded is then $f_{\theta_j}(k_j) \in \mathbb{R}^D$, where $D$ is the embedding dimension. The $M$ metadata vectors are then added together $\boldsymbol{m} = f_{\theta_1}(k_1) + \cdots + f_{\theta_M}(k_M)$, where $\boldsymbol{m} \in \mathbb{R}^D$, which is then also added with with the embedded timestep $\boldsymbol{t} = f_\theta(t) \in \mathbb{R}^D$, so that the final conditioning vector is $\boldsymbol{c} = \boldsymbol{m} + \boldsymbol{t}$.

To summarize, we first we first encode an image $\mathbf{x} \in \mathbb{R}^{C \times H \times W}$ using the SD variational autoencoder (VAE) (Rombach et al., 2022; Esser et al., 2021) to a latent representation $\mathbf{z} = \mathcal{E}(\mathbf{x}) \in \mathbb{R}^{C' \times H' \times W'}$. Gaussian noise is then added to the latent image features to give us $\mathbf{z}_t = \alpha_t \mathbf{z} + \sigma_t \epsilon$ (see section 2). The conditioning vector $\boldsymbol{c}$, created from embedding metadata and the diffusion timestep, as well as the CLIP-embedded text caption $\tau' = \mathcal{T}_\theta(\tau)$, are passed through a DM $\epsilon_\theta(\mathbf{z}_t; \tau', \boldsymbol{c})$ to predict the added noise. Finally, the VAE decoder $\mathcal{D}$ upsamples the denoised latents to full resolution (fig. 1).

Lastly, we initialize the encoder $\mathcal{E}$, the decoder $\mathcal{D}$, the CLIP text encoder $\mathcal{T}_\theta$, and the denoising unet $\epsilon_\theta$ all with SD 2.1's weights. We only update the denoising UNet $\epsilon_\theta$ and the metadata and timestep embeddings $f_{\theta_j}$ during training to speed up convergence using the rich semantic information in the pretrained SD weights. During training, we also randomly zero out the metadata vector $\boldsymbol{m}$ with a probability of 0.1 to allow the model to generate images when metadata might be unavailable or inaccurate. A similar strategy is employed to learn unconditional generation by Ho et al. (2020).

**Single Image-Text-Metadata Datasets** There is no equivalent of a large, text-image dataset (eg: LAION (Schuhmann et al., 2022)) for satellite images. Instead, we compile publicly available annotated satellite data and contribute a large, high-resolution generative dataset for satellite images. Detailed descriptions on how the caption is generated for each dataset are in the appendix. (i) **fMoW**: Function Map of the World (fMoW) Christie et al. (2018) consists of global, high-resolution (GSD 0.3m-1.5m) MAXAR satellite images, each belonging to one of 62 categories. We crop each image to 512x512 pixels. The metadata we consider include longitude, latitude, GSD (in meters), cloud

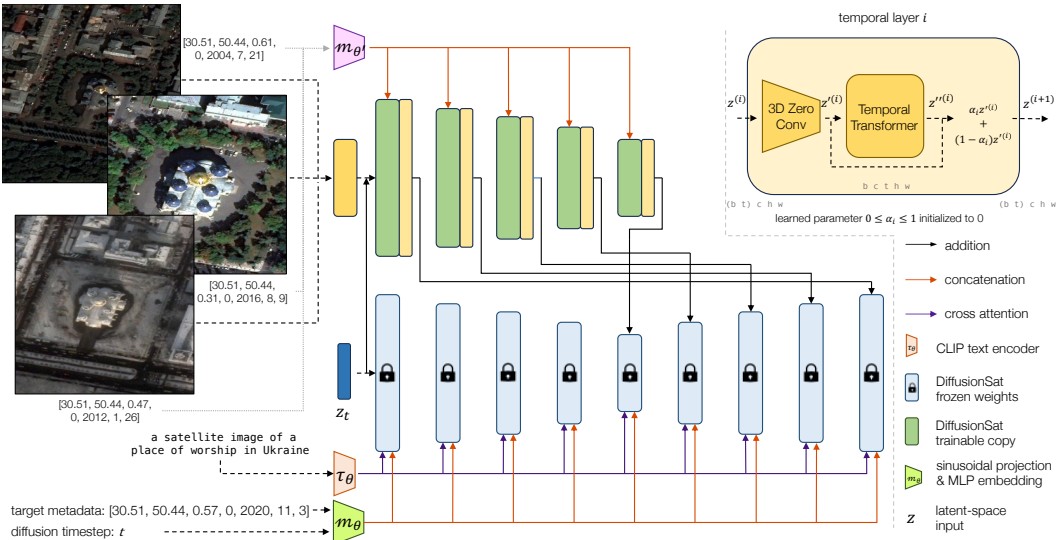

Figure 2: DiffusionSat flexibly extends to a variety of conditional generation tasks. We design a 3D version of a ControlNet (Zhang & Agrawala, 2023) which can accept a sequence of images. Like regular ControlNets, our 3D ControlNet keeps a trainable copy of SD weights for the downsampling and middle blocks. Latent image features are reshaped to combine the batch and temporal dimensions before being input to these layers. The output of each SD block is then passed through a temporal layer (top right), which re-expands the temporal dimension before passing the latent features though a 3D convolution (initialized with zeros) and a temporal, pixel-wise transformer. The metadata associated with each input image is projected as in fig. 1.

cover (as a fraction), year, month, and day. To generate a caption, we consider the semantic class and the country code. (ii) **Satlas**: Satlas Bastani et al. (2022) is a large-scale, multi-task dataset of NAIP and Sentinel-2 satellite images. For our dataset, we use the NAIP imagery in Satlas-small, roughly of the same size as fMoW. We use the same metadata as in item (i). (iii) **SpaceNet**: Spacenet Van Etten et al. (2018; 2021) is a collection of satellite image datasets for tasks including object detection, semantic segmentation and road network mapping. We consider a subset of Spacenet datasets, namely Spacenet v1, Spacenet v2, and Spacenet v5. We use the same metadata as earlier.

## 3.2 CONTROL SIGNAL CONDITIONAL GENERATION

Single-image DiffusionSat can generate a high-resolution satellite image given associated prompt and metadata, but it cannot yet solve the inverse problems described in section 1. To leverage its pretrained weights, we can use it as a prior for conditional generation tasks which do encompass inverse problems such as super-resolution and in-painting. Thus, we now consider generative tasks where we can additionally condition on control signals (eg: sequences of satellite images) $\mathbf{s} \in \mathbb{R}^{T \times C' \times H' \times W'}$, with associated metadata $\mathbf{k}_s \in \mathbb{R}^{T \times M}$, a single caption $\tau$ and target metadata $\mathbf{k} \in \mathbb{R}^M$. Here, $C'$, $H'$, and $W'$ reflect the possible difference in the number of channels, height, and width, respectively, between the conditioning images and the target image. The goal is to sample $\tilde{\mathbf{x}} \sim p(\cdot | \mathbf{s}; \mathbf{k}_s; \tau; \mathbf{k})$, where $\tilde{\mathbf{x}}$ is a sample conditioned on the control signal $\mathbf{s}$ for a given caption $\tau$ and given metadata $\mathbf{k}$.

**Temporal Generation** Recent works for video diffusion have proposed using 3D convolutions and temporal attention (Blattmann et al., 2023; Wu et al., 2022; Zhou et al., 2022), while others propose using existing 2D UNets and concatenating temporal frames in the channel dimension (Voleti et al., 2022; An et al., 2023). However, sequences of satellite images differ in a few key ways from frames of images in a video: (i) there is high variance in the length of time separating images in the sequence, while frames in video data are usually separated by a fixed amount of time (fixed frame rate) (ii) the length of time between images can be on the order of months or years, therefore capturing a wider range of semantic information than consecutively placed frames in video (eg: season, human development, land cover). (iii) there is a sense of "global time" across locations. Even if one compares satellite images across different countries or terrains, patterns may be similar if the

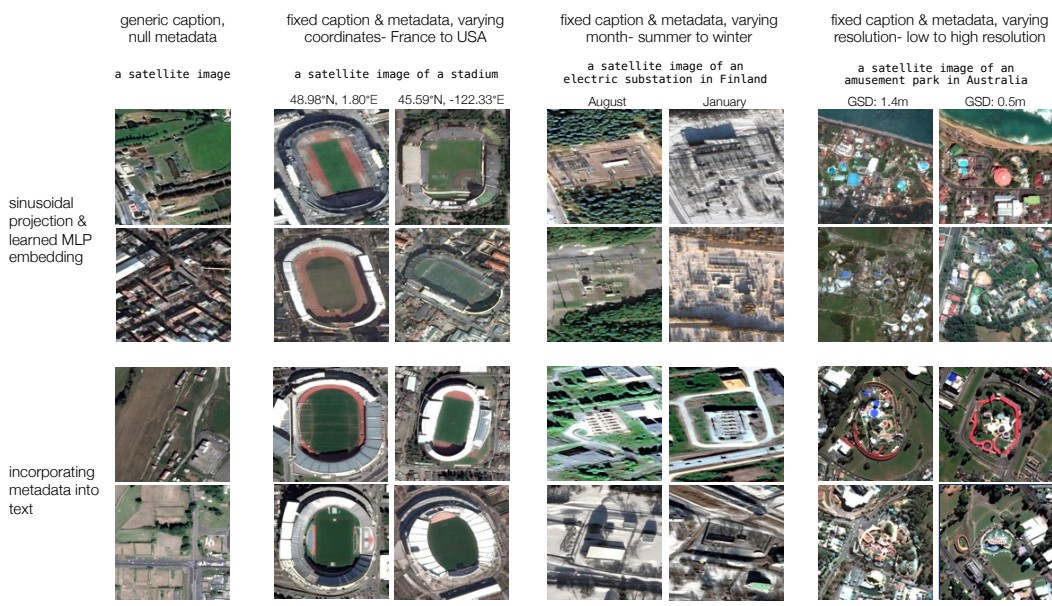

Figure 3: Here we generate samples from single-image DiffusionSat. We see that changing the coordinates from a location in Paris to one in USA changes the type of stadium generated, with American football and baseball more likely to appear in the latter location. Additionally, for locations that receive snow, DiffusionSat accurately captures the correlation between location and season. However, naively incorporating the metadata into the text caption results in poorer conditioning flexibility across geography and season, (eg: with winter and summer time images produced for both August and January, or a lack of "zooming in" when lowering the GSD).

year is known to be 2012 as opposed to 2020 (especially for urban landscapes). This is not the case for video data, where "local" time across frames is sufficient to provide semantic meaning.

Usually, sequences of satellite images have fewer images than frames in natural image videos. As such, generating long sequences of images is less useful than conditioning on existing satellite imagery to predict the future or interpolate in the past (He et al., 2021; Bastani et al., 2023). Thus, we introduce our novel conditioning framework shown in fig. 2 to solve the inverse problem of frame-by-frame conditional temporal prediction. Unlike 2D ControlNet, we use 3D zero-convolutions between each StableDiffusion block (Zhang & Agrawala, 2023). Our temporal attention layers, similar to VideoLDM (Blattmann et al., 2023), further enable the model to condition on temporal control signals. We introduce a learned parameter $\alpha_i$ for each block $i$ to "mix" in the output of the temporal attention layer to prevent noise from early stages in training from affecting our pre-trained weights (fig. 2).

A key advantage of our approach is the ability to provide each item in the control sequence $s$ with its own associated metadata, which is done similarly as in fig. 1, by projecting each metadatum individually and embedding it with an MLP. The embedded metadata for each image is then concatenated with its image and passed through the 2D layers of the ControlNet. DiffusionSat is thus invariant to the ordering of images in the control sequence $s$, since each image's temporal position is solely determined by the timestamp in its metadata. A single DiffusionSat model can then be trained to predict images in the past and future, or interpolate within the temporal range of the sequence.

**Super-resolution with multi-spectral input**   Unlike in section 3.2, our input is a sequence $s$ of lower resolution (GSD) images than the target image and can contain a differing number of channels. The output of the model is still a high-resolution RGB image, as before.

**Temporal Inpainting**   The task is functionally equivalent to 3.2, except the goal is to in-paint corrupted pixels (eg: from cloud cover, flooding, fire-damage) rather than *predict* a new frame in s.

## 4   EXPERIMENTS

We describe the experiments for the tasks in section 3. Implementation details are in appendix A.1.

| Method | FID↓ | IS↑ | CLIP↑ |
|---|---|---|---|
| SD 2.1 | 117.74 | 6.42 | 17.23 |
| SD 2.1† | 37.99 | 7.42 | 16.59 |
| SD 2.1 ‡ | 24.23 | **7.60** | **18.62** |
| Ours | **15.80** | 6.69 | 17.20 |

Table 1: Single-image 512x512 generation on the validation set of fMoW. † refers to finetuned SD 2.1 without any metadata information. ‡ refers to incorporating the metadata in the text caption.

| Method | SSIM↑ | PSNR↑ | LPIPS↓ | MSE↓ |
|---|---|---|---|---|
| Pix2Pix | 0.1374 | 8.2722 | 0.6895 | 0.1492 |
| DBPN | 0.1518 | **11.8568** | 0.6826 | **0.0680** |
| SD | 0.1671 | 10.2417 | 0.6403 | 0.0962 |
| SD + CN | 0.1626 | 10.0098 | 0.6506 | 0.1009 |
| Ours | **0.1703** | 10.3924 | **0.6221** | 0.0928 |

Table 2: Image sample quality quantitative results on fMoW superresolution. DBPN refers to Haris et al. (2018), Pix2Pix is from Isola et al. (2017). Our method beats other super-resolution models for multi-spectral data.

For single image generation, we report standard visual-quality metrics such as FID (Heusel et al., 2017), Inception Score (IS), and CLIP-score (Radford et al., 2021). For conditional generation, given a reference ground-truth image, we report pixel-quality metrics including SSIM (Wang et al., 2004), PSNR, LPIPS (Zhang et al., 2018) with VGG (Simonyan & Zisserman, 2014) features. As noted in Gong et al. (2021) and He et al. (2021), LPIPs is a more relevant perceptual quality metric used in evaluating satellite images. Our metrics are reported on a sample size of 10,000 images.

## 4.1 SINGLE IMAGE GENERATION

We first consider single-image generation, as the task that DiffusionSat is pre-trained on. We compare against a pre-trained SD 2.1 model Rombach et al. (2022), a SD 2.1 model finetuned on our dataset with our captions, but without metadata, and finally a SD 2.1 model finetuned on our dataset with the metadata included in the caption (see table 1). We find that including the metadata, even within the caption, is better than a caption formed from just the labels of satellite images. This is reflected in better FID scores, which measure visual quality. We expect that the text-metadata model ‡ does better in terms of CLIP score given its more highly descriptive caption. However, treating metadata numerically, as in DiffusionSat, further improves generation quality and control, as seen in fig. 3.

## 4.2 CONTROL SIGNAL CONDITIONAL GENERATION

We now use single-image DiffusionSat as an effective prior for the conditional generation tasks of super-resolution, temporal generation/prediction, and in-painting. We describe the dataset for each task and demonstrate results using our 3D conditioning apporach on Texas-housing super-resolution, fMoW super-resolution using fMoW-Sentinel multispectral inputs, temporal generation on the fMoW-temporal dataset, and temporal inpainting on the xBD natural disaster dataset. DiffusionSat achieves state of the art LPIPs and close to optimal performance on the SSIM and PSNR metrics as well.

**fMoW Superresolution** Using the dataset provided in Cong et al. (2022), we create a fMoW-Sentinel-fMoW-RGB dataset with paired Sentinel-2 (10m-60m GSD) and fMoW (0.3-1.5m GSD) images at each of the original fMoW-RGB locations. Given all 13 multi-spectral bands of the Sentinel-2 image (here $T = 1$), we aim to reconstruct the corresponding high resolution RGB image. Super-resolution (section 4.2) given low-resolution (10m-60m), multi-spectral input is especially difficult, since most fMoW-RGB images are <1m GSD. We find that DiffusionSat once again outperforms strong super-resolution baselines, such as SD (which has shown to (table 2, fig. 4). We further note that while methods such as DBPN (Haris et al., 2018) yield strong PSNR/SSIM, these metrics don't reflect human perception and favor blurriness over sharp detail (Zhang et al., 2018; Saharia et al., 2022b).

**Texas Housing Superresolution** The dataset for this task is introduced by Spatial Temporal Superresolution (STSR) (He et al., 2021) and contains 286717 houses built between 2014 and 2017 in Texas. Each location consists of two high-resolution images from NAIP (GSD 1m) and 2 low-resolution images from Sentinel-2 (GSD 10m). A high resolution image at a time $t$ and corresponding low-resolution images at times $t$ and $t'$ form the control signal $s$, and the task is to reconstruct the other high resolution image $x$ at time $t'$.

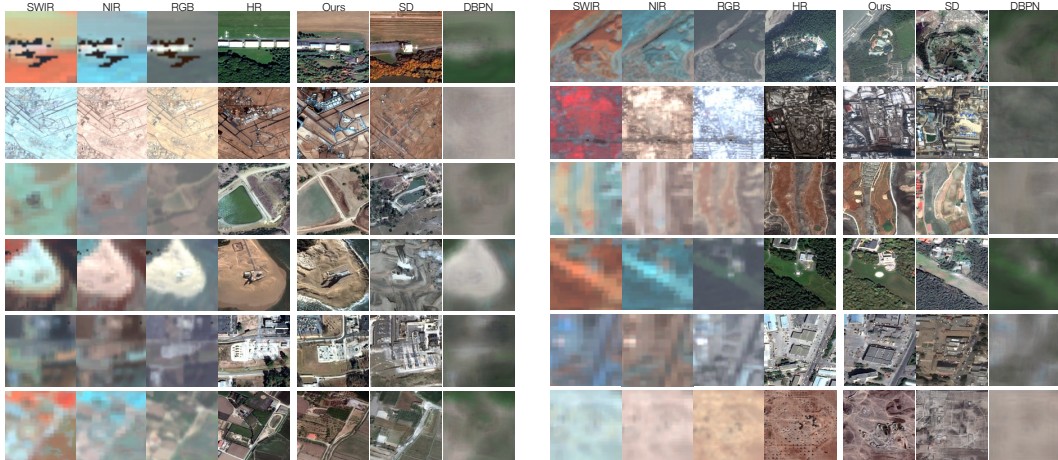

Figure 4: Generated samples from fMoW-Sentinel superresolution validation set. The conditioning image is the Sentinel-2 multispectral (MS) image represented here as SWIR, NIR, RGB. The desired output is the high-resolution (HR) fMoW-RGB image. Our method is able to capture fine-grained details better than other baselines, even when the low-resolution MS image lacks detail. SD tends to "hallucinate" details.

We also perform an ablation on the efficacy of pretraining on our single image datasets against finetuning directly on SD weights. We find a significant improvement from DiffusionSat pretraining, and and from using the 3D ControlNet (across all metrics) over simply stacking the images in the channel dimension and using a 2D ControlNet (table 3).

| Model | $t' > t$ | | | $t' < t$ | | |
|---|---|---|---|---|---|---|
| | SSIM↑ | PSNR↑ | LPIPS↓ | SSIM↑ | PSNR↑ | LPIPS↓ |
| Pix2Pix | 0.5432 | 20.8420 | 0.4243 | 0.3909 | 17.9528 | 0.4909 |
| cGAN Fusion | 0.5976 | 21.5226 | 0.3936 | 0.4220 | 17.8763 | 0.4726 |
| DBPN | 0.5781 | 21.4716 | 0.5101 | 0.4572 | 18.9330 | 0.5910 |
| SRGAN | 0.5361 | 21.1968 | 0.5261 | 0.4221 | 18.9772 | 0.5694 |
| STSR (EAD) | 0.6470 | 22.4906 | 0.3695 | 0.5225 | 19.7675 | 0.4275 |
| STSR (EA64) | **0.6570** | **22.5552** | 0.3764 | **0.5338** | **19.8547** | 0.4342 |
| SD + 3D ControlNet | 0.4747 | 17.8023 | 0.4166 | 0.3458 | 16.1467 | 0.4351 |
| Ours + ControlNet | 0.5403 | 20.3982 | 0.3874 | 0.4657 | 18.1007 | 0.3652 |
| Ours + 3D ControlNet | 0.5982 | 21.0299 | **0.3247** | 0.4825 | 18.4604 | **0.3534** |

Table 3: Sample quality results on Texas housing validation data. $t' > t$ represents generating an image in the past given a future HR image, and $t' < t$ is the task for generating a future image given a past HR image.

**fMoW Temporal Generation**  Many locations in fMoW (Christie et al., 2018) contain multiple images across time. For our experiments, if $T < 4$, we add copies of the latest image to pad the sequence **s** to 4 images. Given a sequence **s** of conditioning images, DiffusionSat can predict another image at any desired target time by appropriately adjusting the target metadata $\mathbf{k}_s$ (section 3.2) Since prior works aren't designed to predict an image at any given target time, we consider tasks where the target image is chronologically prior to or later than the first image in **s**.

Our experiments show that DiffusionSat outperforms STSR and MCVD (Voleti et al., 2022), as well as regular SD with our 3D ControlNet. Quantitative and qualitative results are in table 4 and fig. 5, respectively. These reveal DiffusionSat's improved ability over the baselines to capture the target date's season (eg: snow, terrain color, crop maturity) as well as development of roads and buildings. Other models, lacking the ability to reason about metadata covariates, often simply copy an input image in the conditioning sequence as their generated output. In A.3.1, we showcase DiffusionSat's novel ability to generate sequences of satellite images without prior conditioning images **s**.

| Model | $t' > t$ | | | $t' < t$ | | |
|---|---|---|---|---|---|---|
| | SSIM↑ | PSNR↑ | LPIPS↓ | SSIM↑ | PSNR↑ | LPIPS↓ |
| STSR (EAD) (He et al., 2021) | 0.3657 | 13.5191 | 0.4898 | 0.3654 | 13.7425 | 0.4940 |
| MCVD (Voleti et al., 2022) | 0.3110 | 9.6330 | 0.6058 | 0.2721 | 9.5559 | 0.6124 |
| SD + 3D CN | 0.2027 | 11.0536 | 0.5523 | 0.2218 | 11.3094 | 0.5342 |
| DiffusionSat + CN | 0.3297 | 13.6938 | 0.5062 | 0.2862 | 12.4990 | 0.5307 |
| DiffusionSat + 3D CN | **0.3983** | **13.7886** | **0.4304** | **0.4293** | **14.8699** | **0.3937** |

Table 4: Sample quality quantitative results on fMoW-temporal validation data. $t' > t$ represents generating an image in the past given a future image, and $t' < t$ is the task for generating a future image given a past image.

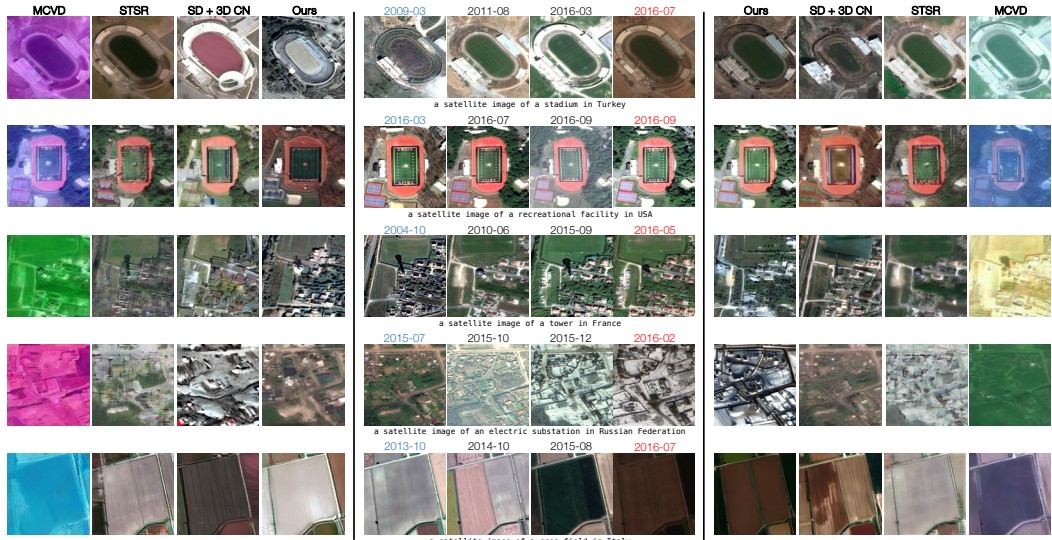

Figure 5: Generated samples from the fMoW-temporal validation set, for temporal prediction. The 4 columns in the center are ground-truth images from the temporal sequence. To the right, we see generated samples for the future-prediction task. The goal is to generate the image marked by the date in red, given the 3 other images (to its left) as conditioning signals. Similarly, for the past-prediction task on the left, the goal is to predict the image marked by the date in blue given the 3 images to its right. DiffusionSat leverages pretrained weights to capture seasonal changes and predict human development better than the baselines. Images are best viewed zoomed in.

**In-painting**   Rather than artificially corrupt input images, we use the xBD dataset (Gupta et al., 2019) which is a subset of the xView-2 (Lam et al., 2018) challenge to assess damage caused by natural disasters. Since each location carries a pre- and post-disaster satellite image, we consider the in-painting task of reconstructing damaged areas in the post-disaster image, or introducing destruction to the pre-disaster image, where $T = 1$. We demonstrate qualitative results in fig. 6. DiffusionSat's capability to reconstruct damaged roads and houses for a variety of disasters including floods, wind, fire, earthquakes etc will be important for disaster response teams to identify access routes and assess damage. We also show that DiffusionSat can *add* damage from different natural disasters, which can be useful for forecasting or preparing areas for evacuation.

## 5   RELATED WORK

**Diffusion Models**   Diffusion models (Ho et al., 2020; Song et al., 2020b; Kingma et al., 2021) have recently dominated the field of generative modeling, including application areas such as speech (Kong et al., 2020; Popov et al., 2021), 3D geometry (Xu et al., 2022; Luo & Hu, 2021; Zhou et al., 2021), and graphics (Chan et al., 2023; Poole et al., 2022; Shue et al., 2023). Besides advancements in the theoretical foundation, large-scale variants built on latent space (Rombach et al., 2022; Saharia et al., 2022a; Ho et al., 2022) have arguably been the most influential. With these foundation models came a slew of novel applications such as subject customization (Ruiz et al., 2023; Liu et al., 2023; Kumari et al., 2023) and text-to-3D generation (Poole et al., 2022; Lin et al., 2023; Wang et al., 2023). Many works have also demonstrated these models' impressive adaptation capabilities through

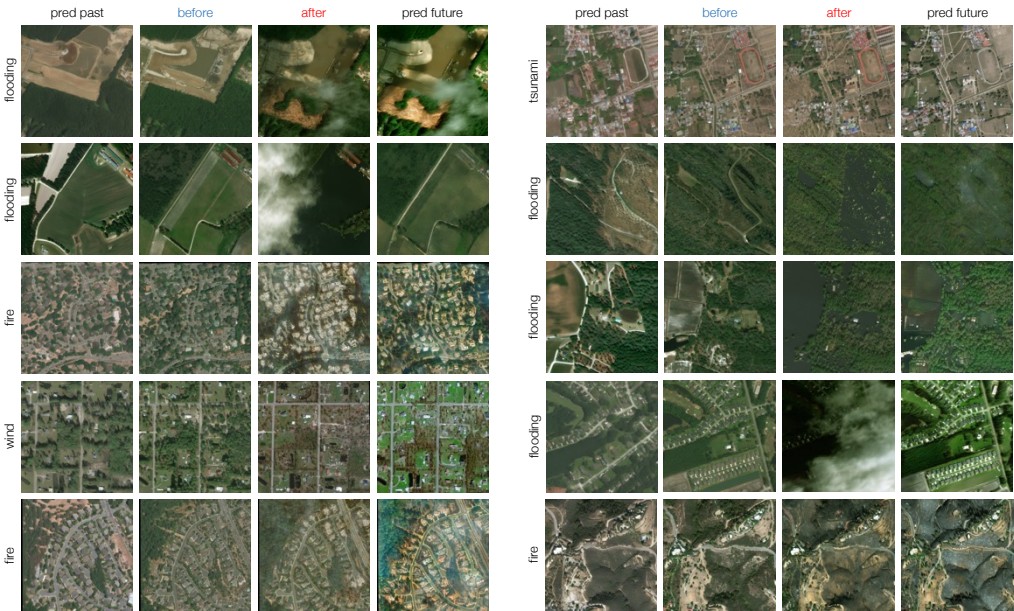

Figure 6: Inpainting results. The two columns marked "before" and "after" represent ground truth images. The "pred past" column is generated by conditioning on the "after" image, and the "pred future" column likewise by conditioning on the "before" image. DiffusionSat successfully reconstructs damaged roads and houses from floods, fires, and wind, even when large portions of the conditioning image are masked by clouds or damage.

finetuning. For example, ControlNet (Zhang & Agrawala, 2023), T2IAdapter (Mou et al., 2023), and InstructPix2Pix (Brooks et al., 2023), which add additional trainable parameters, have proven to be highly successful in adding control signals to the pre-trained diffusion networks.

**Generative Models for Remote Sensing**    Image super-resolution is well studied for natural image datasets (Dong et al., 2015; Ledig et al., 2017; Saharia et al., 2022b; Haris et al., 2018; Rombach et al., 2022). Generative Adversarial Networks (GANs) Goodfellow et al. (2014) such as SR-GAN Ledig et al. (2017) are among the most popular remote-sensing super-resolution methods (Wang et al., 2020; Ma et al., 2019; Gong et al., 2021; Cornebise et al., 2022; Bastani et al., 2023; Rabbi et al., 2020). Other methods have tailor-made convolutional architectures for Sentinel-2 image-input superresolution (Razzak et al., 2023; Tarasiewicz et al., 2023). More recently, Spatial-Temporal Super Resolution (STSR) He et al. (2021) uses a conditional-pixel synthesis approach to condition on a combination of high and low resolution images to generate a high-resolution image at an earlier or later date. In general, these models lack the flexibility and generality of latent-diffusion models across a variety of tasks and datasets, and can suffer from unstable training (Kodali et al., 2017). Our work aims to address these shortcomings by proposing a single approach based on pretrained LDMs that can flexibly translate to downstream generative tasks via our novel conditioning mechanism.

## 6    CONCLUSION

In this work, we provide DiffusionSat, the first *generative* foundation model for remote sensing data based on the latent-diffusion model architecture of StableDiffusion Rombach et al. (2022). Our approach consists of two components: (i) a single-image generation model that can generate high-resolution satellite data conditioned on numerical metadata and text captions (ii) A novel 3D control signal conditioning module that generalizes to inverse problems such as multi-spectral input super-resolution, temporal prediction, and in-painting.

For future work, we would like to explore expanding DiffusionSat to even larger and more diverse satellite imagery datasets. Testing the feasibility of DiffusionSat on generating synthetic data (Le et al., 2023) might also augment existing discriminative methods to scale to larger datasets. Lastly, investigating faster sampling methods or more efficient architectures will enable easier deployment or use of DiffusionSat in resource-constrained settings.

We hope that DiffusionSat spurs future investigation into solving inverse problems posed by remote-sensing data. Doing so would unlock societal benefits to important applications including object detection given super-resolved Sentinel-2 images (Shermeyer & Van Etten, 2019), crop-phenotyping (Zhang et al., 2020), ecological conservation efforts (Boyle et al., 2014; Johansen et al., 2007), natural disaster (eg: landslide) hazard assessment (Nichol et al., 2006), archaeological prospection (Beck et al., 2007), urban planning Li et al. (2019); Xiao et al. (2006); Piyoosh & Ghosh (2017), and precise agricultural applications (Gevaert et al., 2015).

## 7    ACKNOWLEDGEMENTS

This research is based upon work supported in part by the Office of the Director of National Intelligence (ODNI), Intelligence Advanced Research Projects Activity (IARPA), via 2021-2011000004, NSF(#1651565), ARO (W911NF-21-1-0125), ONR (N00014-23-1-2159), CZ Biohub, HAI. The views and conclusions contained herein are those of the authors and should not be interpreted as necessarily representing the official policies, either expressed or implied, of ODNI, IARPA, or the U.S. Government. The U.S. Government is authorized to reproduce and distribute reprints for governmental purposes not-withstanding any copyright annotation therein.

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

# A Appendix

## A.1 Training Details

We list implementation details for our experiments in this section. All models are trained on half-precision and with gradient checkpointing, borrowing from the Diffusers (von Platen et al., 2022) library.

**Single-Image DiffusionSat** We use 8 NVIDIA A100 GPUs. The text-to-image models are trained with a batch size of 128 for 100000 iterations, which we determined was sufficient for convergence. We choose a constant learning rate of 2e-6 with the AdamW optimizer. We train two variants- one for images of resolution 512x512 pixels, and one for 256x256 pixels.

For sampling, we use the DDIM (Song et al., 2020a) sampler with 100 steps and a guidance scale of 1.0. We generate 10000 samples on the validation sets of fMoW-RGB.

**Super-resolution** We use the 512 single-image DiffusionSat model as our prior. We train our ControlNet Zhang & Agrawala (2023) by upsampling the conditional multi-spectral image to 256x256 pixels, which we found to work better than conditioning on 64x64 conditioning images. We use 4 NVIDIA A100 GPUs, and train the model for 50000 iterations with a learning rate of 5e-5 using the AdamW optimizer. We drop Sentinel bands B1, B9, B10, which we find to not be useful, similar to Cong et al. (2022). We use the same sampling configuration as above.

**Texas Housing** We use the 256 single-image DiffuionSat model as our prior. We train our 3D ControlNet on sequences of the HR image and the two LR Sentinel-2 images. We use 4 NVIDIA A100 GPUs, and train the model for 50000 iterations with a learning rate of 5e-5 using the AdamW optimizer. The sampling configuration is the same as above.

**fMoW Temporal** We use the 256 single-image DiffusionSat model as our prior. We train our 3D ControlNet on sequences of at-most 3 conditioning images on the fMoW-temporal dataset. If the location has less than 3 images, we pick one of the conditioning images and copy it over until the sequence is padded to length. We avoid samples where there is only 1 image per location. We train using 4 NVIDIA A100 GPUs, for 40000 iterations with a learning rate of 4e-4 using the AdamW optimzer. The sampling configuration matches the ones above.

## A.2 Datasets

### A.2.1 Captions and metadata

The text captions are dependent on the metadata fields available for each dataset. For the captions below, fields denoted in angle brackets below are filled in using the metadata for each example. Some sections of each caption, denoted by square brackets below, are randomly and independently dropped out of caption instances at a 10% rate. We label datasets from the same satellite sources with the same image type (e.g., both Texas Housing and Satlas use NAIP images, so both are labelled as `satlas` images).

| Dataset | Caption |
|---|---|
| fMoW | `"a [fmow] satellite image [of a <object>] [in <country>]"` |
| SpaceNet | `"a [spacenet] satellite image [of <object>] [in <city>]"` |
| Satlas | `"a [satlas] satellite image [of <object>]"` |
| Texas Housing | `"a [satlas] satellite image [of houses] [built in <year_built>] [covering <num_acres> acres]"` |
| xBD | `"a [fmow] satellite image [<before/after>] being affected by a <disaster_type> natural disaster"` |

Table 5: Captions created for each dataset type based on available label information.

Besides the captions, we also incorporate numerical metadata from 7 fields. Each field was normalized based on high and low reference values: $m_{norm} = m/(high - low) \times scale$, where $scale$ is a scaling

| field | description | min | max |
|---|---|---|---|
| lon | longitude, in degrees | -180 | 180 |
| lat | longitude, in degrees | -90 | 90 |
| gsd | ground sampling distance | 0 | 10 |
| cloud_cover | proportion of pixels with cloud cover | 0 | 1 |
| year | year of the satellite image | 1980 | 2100 |
| month | month of the year | 0 | 12 |
| day | day of the month | 0 | 31 |

| Dataset | Image | Caption | Metadata |
|---|---|---|---|
| fmow |  | a fmow satellite image of a car dealership in United States of America | lon: -76.781
lat: 17.98
gsd: 0.941
cloud_cover: 0
year: 2010
month: 10
day: 6 |
| satlas |  | a satlas satellite image of 26 ms buildings | lon: 78.995
lat: 85.048
gsd: 2
cloud_cover: 0
year: 2013
month: 6
day: 22 |
| spacenet |  | a spacenet satellite image of 144 buildings covering an area of 9280.166 squared meters in Rio | lon: -43.636
lat: -22.892
gsd: 0.793
cloud_cover: 0
year: 1980
month: 0
day: 0 |

Figure 8: Sample captions and pre-normalization metadata for the fMoW, Satlas, and SpaceNet datasets.

factor of 1000, such that $low$ maps to 0 and $high$ maps to $scale$. The fields are summarized in Figure 7. We include examples of both the captions and numerical metadata in Figure 8.

## A.3 TEMPORAL GENERATION

In this section, we provide further results for the temporal generation task, demonstrating the powerful capabilities of DiffusionSat.

### A.3.1 SEQUENCE GENERATION

We first demonstrate how we can generate temporal sequences of satellite images *unconditionally* i.e. without any prior conditioning image, unlike in section 4.2. To do so, we first generate a satellite image using single-image DiffusionSat given a caption and desired metadata for our image. We then apply our novel 3D-conditioning ControlNet, already trained for temporal generation, on the first image to generate the next image in the sequence, given some desired metadata (eg: how many years/months/days into the future or past). We now re-apply the 3D-conditioning ControlNet on the first 2 generated images to get the third image of the sequence. Repeating this procedure, we are

able to auto-regressively sample sequences of satellite images given desired metadata properties. Generated samples using this procedure are shown in fig. 9.

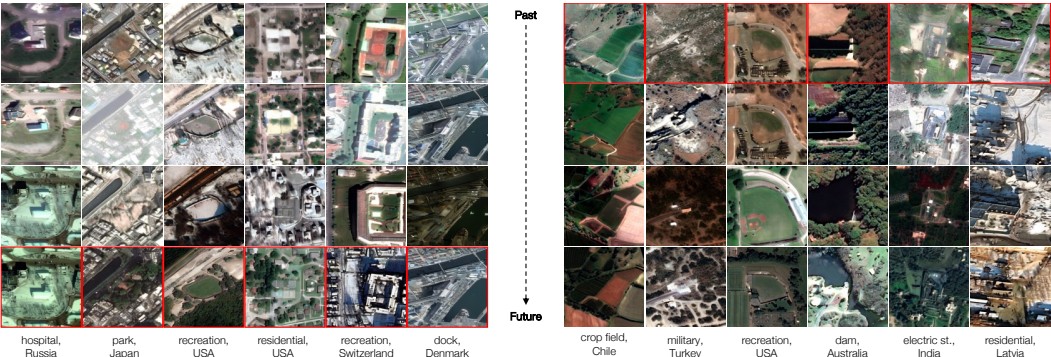

hospital, Russia | park, Japan | recreation, USA | residential, USA | recreation, Switzerland | dock, Denmark

crop field, Chile | military, Turkey | recreation, USA | dam, Australia | electric st., India | residential, Latvia

Figure 9: Auto-regressively generated sequences of satellite images given a caption (for the sequence) and desired metadata (per image). The image sampled from single-image DiffusionSat is outlined in red. On the left, we generate sequences backwards in time (i.e.: into the past) given the first generated image. For example, for the park in Japan (2nd column from the left), the metadata, from the bottom to the top image, is: (1.07, 2014, 5, 25), (1.62, 2013, 6, 17), (1.17, 2011, 3, 17), (1.17, 2010, 11, 8). The metadata is in order (GSD, year, month, day). We omit listing the latitude and longitude, since that remains the same for the sequence, but it is inputted as metadata, as described in fig. 1. On the right, we generate images forwards in time (i.e. into the future) given the first generated image outlined in red. For example, for the crop field in Chile, the metadata, from the bottom to the top image, is: (1.03, 2016, 2, 23), (1.03, 2015, 9, 17), (0.97, 2014, 10, 12), (1.17, 2014, 8, 13). As we can see, our model generates realistic sequences that reflect trends in detail and development both forwards and backwards in time.

Our results demonstrate a novel way of generating arbitrarily long sequences of satellite images- our conditioning mechanism can flexibly handle both conditional and unconditional generation. The generated samples reflect season and trends in development (eg: past images have fewer structures, future images usually have more detail).

## A.4   GEOGRAPHICAL BIAS

Concerns about bias for the outputs of machine learning models are natural given the large, potentially biased datasets they are trained on (Huang et al., 2021). We perform an evaluation of the generation quality of single-image DiffusionSat across latitude and longitude around the globe in fig. 10 and fig. 11

Our results show no particular favoritism for location, even though one would expect better generation quality for regions in North America and Europe. We would still like to point out a few caveats:

(i) FID or LPIPs scores may not be the most informative metric towards estimating bias in sample quality. We use it as a measure of generation quality given a lack of better alternatives for the novel problem of estimating geographical bias in generative remote sensing models.

(ii) The FID scores are dependent on sample size, and so while the scores might be evenly distributed, it still remains the case that there are far more dataset samples from developed regions of the world, and a dearth of images for large swaths (eg: across Africa). Even so, for a severely biased model we would expect poorer generation quality for data-poor regions of the world.

(iii) We estimate only one angle of bias. Bias may still exist along different axes, such as generating types of buildings, roads, trees, crops, and understanding the effects of season. We leave this investigation to future work.

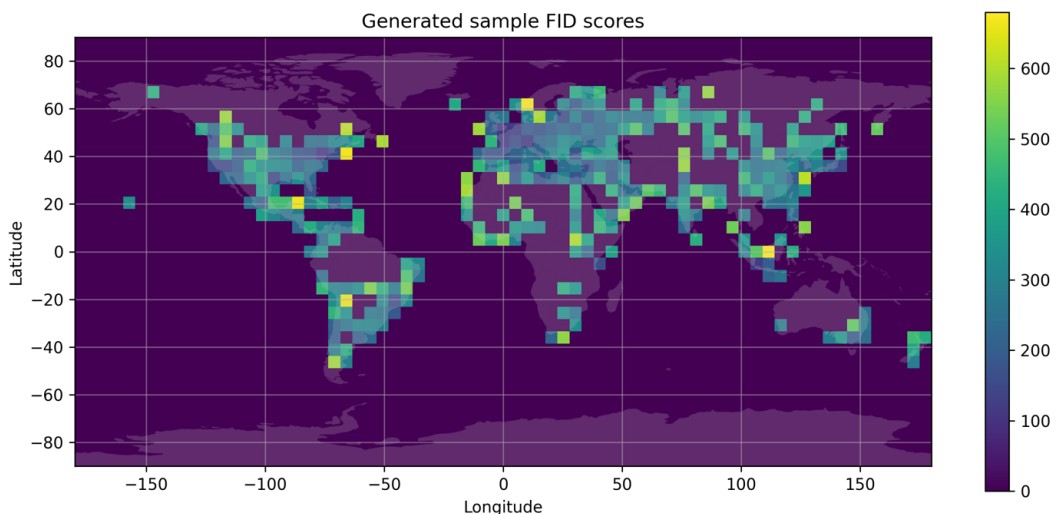

Figure 10: FID scores of single-image DiffusionSat prompted on 10k samples of the fMoW-RGB validation set for coordinates around the world.

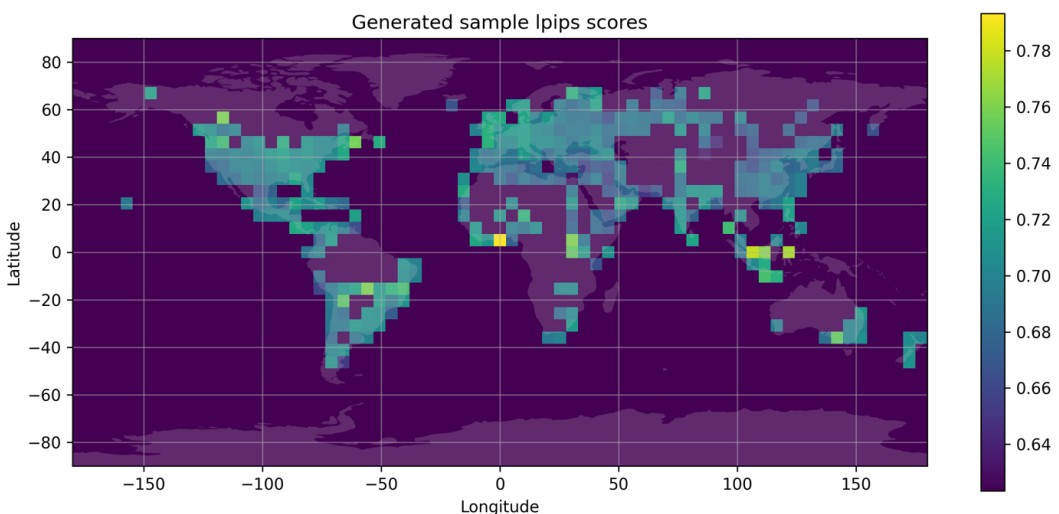

Figure 11: LPIPs scores of super-resolution DiffusionSat prompted on 10k samples of the fMoW-Sentinel-fMoW-RGB validation set for coordinates around the world.

