# OpenReview forum: "DiffusionSat: A Generative Foundation Model for Satellite Imagery"
_ICLR.cc/2024/Conference — ICLR 2024 poster_

### Official Review · Reviewer_tt9n · 2023-10-31

**Soundness:** 4 excellent
**Presentation:** 4 excellent
**Contribution:** 3 good
**Rating:** 8
**Confidence:** 4

**Summary:**

The authors propose a stable diffusion variant, called DiffusionSat, designed for satellite image synthesis tasks. The proposed approach leverages text captions, an assortment of satellite image metadata (longitude, latitude, GSD, etc.), and sequences of images to address various overhead image problems, such as varying spatial resolution (GSD), cloud cover, and seasonality.  DiffusionSat shows strong performance across multiple tasks (super-resolution, temporal generation, and in-painting) on three repurposed datasets.

**Strengths:**

- Tackles an important and interesting problem, view synthesis in remote sensing.

- Well-written, clear motivation.

- Method has the potential to positively impact the remote sensing community.

  DiffusionSat operates on a sequence of satellite images (with varying spatial resolution) and applies attention across time via a temporal pixel-transformer. This allows the approach to leverage an image sequence to address the shortcomings that may be present in any individual image (e.g., clouds). While the underlying temporal transformer was introduced by VideoLDM (Blattmann et al., 2023), its application to remote sensing problems is still valuable.  Further, the proposed method allows satellite image synthesis to be controlled using metadata and text. For example, synthesizing images with seasonality (summer grass vs winter snow), location (France vs.  USA), and ground sample distance.

- The authors introduce a new temporal generation task using satellite image datasets. This task targets common issues with using satellite image sequences, namely that the interval between images is not fixed and that frames can have varying GSD.

- Evaluation is fairly extensive.

**Weaknesses:**

- Limited technical novelty.

  The proposed method reduces to a straightforward way of embedding metadata and then conditioning a stable diffusion model on said metadata. The temporal transformer component is from VideoLDM.

- Quantitative performance across multiple tasks is strong, but not convincingly better than existing approaches. While the proposed method does achieve SOTA performance on some metrics on some benchmarks, it doesn't show consistent enough performance across the board to substantiate the claim that the method is SOTA on all these tasks, as the introduction indicates.

- The approach and evaluation are limited for multi-spectral imagery (MSI).

  This work does not address the issue of varying image band GSD (e.g., Sentinel-2 bands) in MSI, despite claiming to support MSI synthesis. For various remote sensing products, image bands may not all be the same GSD. Some effort should be taken to recognize this occurrence, since 10m -> 1.5m super-resolution is far easier than 20m->1.5m (Sentinel-2 bands 5, 6, 7, 8a, 11, and 12) or 60m -> 1.5m (Sentinel-2 bands 1, 9, and 10) [1] super resolution.

  [1] https://sentinels.copernicus.eu/web/sentinel/user-guides/sentinel-2-msi/resolutions/spatial

  The MSI super-resolution problem aims to synthesize 3-band WV3 imagery from 13-band Sentinel-2 imagery. While this may be useful in some scenarios, it does leave out a fairly significantly problem, namely, generating all spectral bands. As such, the method does not really perform MSI synthesis, but rather can support MSI inputs for RGB super-resolution.

- Figure 4 is not informative as the disaster-related damage is difficult to identify. Consider highlighting damaged areas, or use a more clear example.

- Minor editing problems: leftover question mark (page 4), repetitive text (bottom of pages 1 and 3), typo and highlighting on page 8, etc.

**Questions:**

My initial rating is weak accept. The proposed approach is interesting and has novelty in its formulation. The manuscript is well written and the results are compelling. Ultimately, I think this paper will have a positive impact on the community.

For the fMoW super-resolution task, are the super-resolved images multi-spectral or just RGB? The text seems to suggest that the model takes in a 13 band Sentinel-2 image and outputs a reconstruction of a 3 band WorldView-3 image from fMoW. Is the reconstruction limited to just 3 bands (R, G, B)? If so, then the task isn't actually multi-spectral super-resolution.

---

> ### Author Response · Authors · 2023-11-20
> **Response to reviewer tt9n**
>
> Thank you for your thoughtful feedback, emphasis of the significance of our problem statement, recognition of DiffusionSat’s advantages over prior works and its novelty in tackling temporal satellite-image generation. We also value your comments on improving the evaluation of our method on different benchmarks and tasks and on clarifying our presentation. You may find responses to your questions below.
>
> **Q: Technical Novelty**
> We would like to stress that part of the novelty lies in identifying the usefulness of metadata towards controlling useful generations from latent diffusion models. Natural images simply use semantic text captions to generate images since these are readily available. We recognize the importance of remote-sensing specific numerical metadata towards generating satellite images that are aligned with season, weather, location, resolution etc. This positively affects not only generation quality, but also allows users to prompt the model in more flexible and fine-grained manner (i.e. with/without detailed text, specific months, resolutions, etc.)
>
> While the temporal transformer is introduced in VideoLDMs, our method uses them as part of a conditioning model. Our temporal controlnet is slightly different from the VideoLDM temporal layers; the 3D convolutions of our controlnet operate by default(i.e. they don’t have the learned mix-in parameter), but the temporal attention layers are mixed-in with a learned parameter. The reason is that we swap out 2D convolutions from the controlnet, and so we require some temporal component to always be active (that is, the 3D convolutions). We did experiment with using the temporal transformer by default, but this didn’t work as well.
>
> We would also argue that using a 3D controlnet in this way is novel, as it unlocks the capabilities of a single-image pretrained LDM (eg: DiffusionSat) to generate arbitrarily long sequences of images auto-regressively. We can do this by either conditioning on a seed sequence of images (eg: real satellite images of a location) and then generate frames in the past, future, or in-between, or generate a completely synthetic sequence of images by sampling the first image of the sequence from single-image DiffusionSat and then using the synthetic image as the seed. In fact, we demonstrate this capability in the appendix, in figure 8, in the revision.
>
> The combined single-image, 3D controlnet approach using numerical metadata allows DiffusionSat to be very general. Control signals of almost any format can be input to our model, whether by numerical metadata or image sequences, to generate a desired output satellite image. This approach to generative modeling for remote sensing data is novel and very flexible.
>
> **Q: Comparison to other baselines**
> Thank you for the suggestion. We have updated our results tables with results from other baseline models. Our results for temporal generation and super-resolution demonstrate DiffusionSat’s improvement over baselines, both in qualitative and quantitative results.
>
> **Q: Quantitative results performance**
> Thank you for your observation. We would like to emphasise that some of the quantitative metrics don’t reveal the full story. Metrics such as SSIM/MSE/PSNR can be optimized by samples that aren’t perceptually realistic [1,2], which can indeed be seen by our generated samples we have included in figures 4 and 5. DBPN doesn’t produce many sharp details or any realistic structures, but is able to achieve a high PSNR/MSE. Our samples demonstrate DiffusionSat’s strong ability to super-resolve on the order of 10-20x and predict the past and future changes to images, including on buildings/construction, crop maturity, season, and location. Given these advancements, we claim that DiffusionSat is state-of-the-art in many generative tasks for remote-sensing.
>
> **Q: Inpainting figure visibility**
> Thank you for the suggestion. We have included a new figure with more detailed generated samples for the in-painting section. Our qualitative results for in-painting demonstrate DiffusionSat’s strong capability to in-paint damaged regions across a variety of disasters. We also demonstrate the ability to introduce damage to images, based on the type of natural disaster. We hope that this tool is useful to researchers and first-responders who wish to understand damaged or obscured structures in a region in the event of a disaster (eg: which roads are flooded/blocked, which houses were razed, how the landscape changes due to disaster etc.)
>
> **Q: Formatting**
> We have included these changes, thank you for catching them!

---

> > ### Author Response · Authors · 2023-11-22
> > **Resolving additional concerns**
> >
> > Dear reviewer,
> > As the discussion deadline approaches, we hope that your concerns are resolved and that you can reconsider our scores. Thank you very much for your time

---

> > ### Comment · Reviewer_tt9n · 2023-11-23
> > **Thanks**
> >
> > Thanks for the response. I will take this additional information into account, as well as the responses to other reviewers, when making my final decision.

---

> ### Author Response · Authors · 2023-11-20
> **Response to reviewer tt9n (continued)**
>
> **Q: Varying image resolution in multi-spectral images**
> Thank you for your suggestion. It is true that sentinel images contain bands of varying resolution. For our super-resolution task, we consider upsampling a sentinel image with bands of GSD 10m, 20m, and 60m to a resolution of around 0.5m-1.5m. We focus on this task as one of high-importance, given the abundance of unlabelled sentinel-2 imagery around the world but the dearth of high-resolution images for multiple locations and time-stamps. Multi-spectral image generation was therefore not one of the tasks we chose to demonstrate, although DiffusionSat can handle this task as well.
>
> We want to emphasize the **generality** of our approach: using our 3D conditioning scheme, we can extend DiffusionSat to a variety of control signals (eg: images, multi-spectral inputs, numerical metadata etc.). This would also include multi-spectral inputs, either for conditional generation, or as its own pretraining dataset for a multi-spectral generative model.
>
> We also would like to mention that our approach would be able to handle multi-spectral inpainting. With a suitable dataset, we can condition on input multi-spectral bands as control signals and super-resolve them to high-resolution. However, we aren’t aware of existing large-scale publicly available datasets which would provide ground truth high resolution spectral bands for multiple different wavelengths. OUr method would be able to handle this task for existing spectral bands in our dataset (eg: RGB). That is, masking parts of one or more input bands, providing the image as conditioning input, and then reconstructing the bands is almost identical to the in-painting tasks we demonstrate.
>
> For varying image resolution in general: we do demonstrate our method’s ability to handle such inputs. For temporal generation, images in a sequence of fMoW images are not necessarily of the same GSD. We input the GSD as metadata precisely so that the model can understand the correlation between GSD and the size of objects in the image. For the texas-housing task, we input one high resolution image (GSD 1m), and two low resolution images (GSD 10m) to form the control sequence **s**. DiffusionSat easily handles conditional generation given this input as it understands the different input metadata per image.
>
> **Q: Multi-spectral super-resolution task definition**
> Thank you for the observation. We should have been clearer. To clarify, the super-resolution task in the paper is one of super-resolving multi-spectral input images to their high-resolution RGB counterparts. With a suitable dataset, DiffusionSat would also be able to generate super-resolved spectral bands given low-resolution conditioning signals, which we leave for future work.

---

> > ### Author Response · Authors · 2023-11-23
> > **References accompanying response to reviewer tt9n**
> >
> > [1] Blau, Yochai, and Tomer Michaeli. "The perception-distortion tradeoff." In Proceedings of the IEEE conference on computer vision and pattern recognition, pp. 6228-6237. 2018.
> >
> > [2] Zhang, Richard, Phillip Isola, Alexei A. Efros, Eli Shechtman, and Oliver Wang. "The unreasonable effectiveness of deep features as a perceptual metric." In Proceedings of the IEEE conference on computer vision and pattern recognition, pp. 586-595. 2018.

---

> ### Author Response · Authors · 2023-11-23
> **Resolving additional concerns**
>
> Dear reviewer,
>
> Thank you for your response, we appreciate your thoughtful feedback and discussion. We hope that your further concerns are resolved and can reconsider our scores.
>
> Thanks!

---

### Official Review · Reviewer_DCYE · 2023-10-31

**Soundness:** 3 good
**Presentation:** 3 good
**Contribution:** 3 good
**Rating:** 6
**Confidence:** 4

**Summary:**

Authors propose DiffusionSat which is a Stable Diffusion (SD) inspired generative model to generate satellite imagery. The imagery generation can be conditioned satellite imagery metadata. To be able to train this model authors had to generate a large-scale satellite imagery pretraining dataset by combining multiple open datasets, generating text descriptions, and adding metadata available. Work was done to encode the conditioning metadata properly. The pretrained model was tested in multiple downstream tasks we competitive performance.

**Strengths:**

* This paper tackles an important problem with understady on the computer vision field with impotant possitive societal benefiting applications
* Novel incorporation of additional metadata and problem setup and can spin off a new line of work in geospatial ML
* The generated dataset used for this study can be very useful in other applications.

**Weaknesses:**

* **Results for certain tasks are missing or incomplete.** The paper mentions that they show state-of-the-art results for super-resolution, temporal generation, and in-painting. However, only a single qualitative example is provided as result. Also multiple other relevant approaches have been proposed. Superresolution results just compare the proposed approach with Stable Diffusion baseline but ignores the line of work done in the field including [1,2] and others.
* **The paper write up needs work to improve cohesion.** The related work section could be merged into the background. The task description coud be joint with the results since bothe are very brief.
* **Overall evaluation and results comparison.** There are multiple other approaches for satellite image generation including conditional that this approach does not compare against. [1]  If the focus wants to be on using this model as foundational model for other geospatial ml tasks, then comparison with other self-supervised and/or foundational models is important.


References:
1. Tarasiewicz, Tomasz, et al. "Multitemporal and multispectral data fusion for super-resolution of Sentinel-2 images." IEEE Transactions on Geoscience and Remote Sensing (2023).
2. Razzak, Muhammed T., et al. "Multi-spectral multi-image super-resolution of Sentinel-2 with radiometric consistency losses and its effect on building delineation." ISPRS Journal of Photogrammetry and Remote Sensing 195 (2023): 1-13.
3. Le, V. A., Reddy, V., Chen, Z., Li, M., Tang, X., Ortiz, A., ... & Robinson, C. (2023). Mask Conditional Synthetic Satellite Imagery. arXiv preprint arXiv:2302.04305.

**Questions:**

1. It is very difficult to see the inpainted damage referred to in Figure 4. Please consider highlighting the area where damage has been inpainted.
2. Abstract missing dot after "remote sensing datasets"
3. No text under "In-painting" subsection within the experiments section.
4. Share more examples of generated images

---

> ### Author Response · Authors · 2023-11-20
> **Response to reviewer DCYE**
>
> Thank you for your constructive feedback, recognition of the novelty of our approach and the significance of our problem statement, and the wider relevance of our benchmark evaluations and datasets. We also appreciate your feedback to improve the presentation and clarity of our experimental section, and to compare our work against other relevant baselines. You may find responses to your questions below:
>
> **Q: Missing results**
> Thank you for your observation. We have added results comparing DiffusionSat with other baselines to the super-resolution and temporal generation tasks. We demonstrate an improvement over prior state-of-the-art super-resolution models including StableDiffusion [1], DBPN [2], and Pix2Pix [3]. We also include qualitative samples for the super-resolution task. These results demonstrate DiffusionSat’s strong capability to super-resolve very low resolution input images to reveal broad structures and landscape. Other methods typically struggle with very low resolution inputs, as can be seen with DBPN producing blurry images. StableDiffusion produces realistic-looking images, but more frequently hallucinates details.
>
> For temporal generation, we include Masked Conditional Video Generation (MCVD) a state-of-the-art conditional video generation diffusion model. Our results demonstrate an improvement over this method, as well as over StableDiffusion adapted with out 3D controlnet. Qualitative samples reflect our method’s improvement over prior works, with more realistic images generated both in the past and the future.
>
> We hope that these results further strengthen our case for DiffusionSat and its role as a generative foundation model for remote sensing data.
>
> Thank you also for suggesting other super-resolution baselines. We have cited these works as part of our related work section. However, were unable to locate public code repositories in order to reproduce their method for our paper. We are happy to include the method’s results given pointers to their implementation. We have included other methods with publicly available code to ensure fair comparison with our method and avoid potential bugs from re-implementing previous works from scratch.
>
> **Q: Writing cohesion**
> Thank you for the suggestion. We hope to ease the reader into relevant diffusion-model terminology via the background section, which is relevant as we describe our method in the Method sections. We provide a summary of related work towards the end of the paper to aid the reader between the methods and experiments section for better cohesion.
>
> We have updated our method section to move the description of the datasets used for control signal conditional generation to the experiments section. We have also added further detail in the appendix, including a preliminary investigation into sample quality bias across geography and image-unconditional sequence generation. We hope that these additions improve cohesion.
>
> **Q: Other baseline comparison results and more examples of generated images**
> Thank you for the suggestion. We have also included more baseline comparison results as mentioned above and in the overall response. In addition, we have provided samples for the temporal generation and super-resolution tasks in the paper, comparing the quality of our results with prior works.
>
> In the appendix, we also include a brief section on geographical bias, and samples from our model for unconditional temporal generation (i.e. without conditioning on an input image). We hope that these samples demonstrate the quality of DiffusionSat.
>
> **Q:  Inpainting damage figure limited visibility**
> We have updated this section to show more samples for the in-painting task. Our in-painting section now properly describes our qualitative results on this task. Since the task is very similar to temporal prediction, we defer the reader to refer to those quantitative results for comparison with prior works.
>
> Our qualitative results for in-painting demonstrate DiffusionSat’s strong capability to in-paint damaged regions across a variety of disasters. We also demonstrate the ability to introduce damage to images, based on the type of natural disaster. We hope that this tool is useful to researchers and first-responders who wish to understand damaged or obscured structures in a region in the event of a disaster (eg: which roads are flooded/blocked, which houses were razed, how the landscape changes due to disaster etc.)
>
> **Q: Typos and formatting**
> Thanks for the suggestions, we have taken a closer look and fixed some formatting and typos.

---

> ### Author Response · Authors · 2023-11-20
> **References accompanying response to reviewer DCYE**
>
> [1] Rombach, Robin, Andreas Blattmann, Dominik Lorenz, Patrick Esser, and Björn Ommer. "High-resolution image synthesis with latent diffusion models." In Proceedings of the IEEE/CVF conference on computer vision and pattern recognition, pp. 10684-10695. 2022.
>
> [2] Haris, Muhammad, Gregory Shakhnarovich, and Norimichi Ukita. "Deep back-projection networks for super-resolution." In Proceedings of the IEEE conference on computer vision and pattern recognition, pp. 1664-1673. 2018.
>
> [3] Isola, Phillip, Jun-Yan Zhu, Tinghui Zhou, and Alexei A. Efros. "Image-to-image translation with conditional adversarial networks." In Proceedings of the IEEE conference on computer vision and pattern recognition, pp. 1125-1134. 2017.

---

> ### Author Response · Authors · 2023-11-22
> **Resolving additional concerns**
>
> Dear reviewer,
> As the discussion deadline approaches, we hope that your concerns are resolved and that you can reconsider our scores. Thank you very much for your time

---

### Official Review · Reviewer_ZvAY · 2023-11-01

**Soundness:** 2 fair
**Presentation:** 2 fair
**Contribution:** 1 poor
**Rating:** 3
**Confidence:** 4

**Summary:**

This work proposes a generative foundation model, i.e., DiffusionSat, for remote sensing data based on the latent-diffusion model architecture of StableDiffusion. Conditioning on freely available metadata as well as generated captions on large, publicly available
satellite datasets makes DiffusionSat a powerful and flexible generative model. Further, a novel 3D ControlNet which allows DiffusionSat to generalize to multi-spectral superresolution, temporal prediction, and in-painting is designed.

**Strengths:**

This work proposes a generative foundation model for remote sensing data based on StableDiffusion. The proposed foundation model produces realistic samples and can be used to solve multiple generative tasks including temporal generation, multi-spectral superrresolution and in-painting.

**Weaknesses:**

1. The necessity and motivation of designing the generative foundation model are not clear and convincing.
2. The methodology of training the proposed foundation model is not novel since the whole framework is a combination of stable diffusion and ControlNet.

**Questions:**

1. The proposed foundation model for satellite images is not novel and significant since there are many foundation models in the remote sensing area, such as  [1-2].
[1] A Billion-scale Foundation Model for Remote Sensing Images. arXiv:2304.05215.
[2] Advancing Plain Vision Transformer Towards Remote Sensing Foundation Model. TGRS, 2022.

2. The authors claim that "while foundation models have been recently developed for discriminative learning on satellite images, no such foundation model exists for generative tasks. " Please explain why the existing foundation models are not applicable to the generative tasks.

3. The methodology of training the proposed foundation model is not novel and the authors only apply the stable diffusion model to the satellite images. Although the authors use some new conditional inputs in the stable diffusion framework, it is not novel enough for publication in this conference.

4. What is the purpose of conducting research on the generative foundation model? Please make a discussion for this point.

---

> ### Author Response · Authors · 2023-11-20
> **Response to reviewer ZvAY**
>
> Thank you for your feedback and recognition of DiffusionSat’s capabilities to tackle multiple relevant generative tasks for remote-sensing data. You may find responses to your questions below:
>
> **Q: Motivation for generative foundation models + “What is the purpose of conducting research on the generative foundation model?**
> For natural images, generative foundation models used as priors have led to major improvements in a variety of inverse problems like inpainting, colorization, deblurring [1,2,3], medical image reconstruction [4]. Just as StableDiffusion uses a different architecture and training objective to generate images compared to ViTs that are pre-trained for natural image discriminative tasks (eg: MAE [5], MoCo [6], DINO [7] etc.), we are proposing a different architecture and training objective for remote sensing generative tasks.
>
> Remote sensing data provides unique inverse problems that are different from those in natural images. For example, super-resolving widely available multi-spectral Sentinel-2 enables object detection and classification for regions of the world which lack high-resolution satellite data [8]. High-resolution satellite images are also crucial for crop-phenotyping [9], for conservation efforts in biology [10, 14], and landslide hazard assessment [11], 3D city modelling [12], archaeological prospection [13]. Other inverse problems such as cloud/shadow removal have important uses in urban planning [15,16,17]. Further, temporal generation and multi-spectral imputation have benefits towards precise agricultural applications [18].
>
> These studies demonstrate the real-world importance of solving generative tasks using remote-sensing data for industrial and academic use-cases. However, the challenge is that these papers introduce solutions tailored to solve their specific task, and these solutions cannot generalize to multiple downstream generative tasks in one unified framework. We therefore require more research from the generative perspective for remote sensing and our work is a significant step towards this goal.
>
> **Q: The proposed foundation model for satellite images is not novel and significant since there are many foundation models in the remote sensing area**
> We believe the reviewer has misunderstood the context of our work and has made a hasty judgement brushing off DiffusionSat’s novelty by dismissing the importance of generative models. Neither of the methods the reviewer cited [19, 20] are generative, which further demonstrates the need for such large-scale generative models in this field and novelty of our work.
>
> Claiming that our model is “not novel since there are many foundation models in the remote sensing area” is a rather simplistic view that groups all large-scale models as “foundation models” without noting fundamentally different and important  distinguishing factors, i.e. generative model vs discriminative models. This is unfair to the author, other reviewers, and ACs who will take these hasty judgements into account.
>
> To go into more detail, [19,20] are both large-scale, pretrained vision transformers which can solve downstream discriminative tasks, just like [21] which we mention in our paper as a discriminative foundation model. To ask why they cannot solve generative tasks is akin to asking why a self-supervised method for discriminative tasks such as MoCo [6] cannot generate images from captions like StableDiffusion can. They have fundamentally different theory and design/architecture.
>
> **Q: “Please explain why the existing foundation models are not applicable to the generative tasks.”**
> As mentioned above, the reviewer seems to conflate generative and discriminative models. Discriminative models are not designed to and thus cannot solve any of the inverse problems we mention, including super-resolution, cloud/shadow removal, temporal interpolation.

---

> ### Author Response · Authors · 2023-11-22
> **Resolving additional concerns**
>
> Dear reviewer,
> As the discussion deadline approaches, we hope that your concerns are resolved and that you can reconsider our scores. Thank you very much for your time

---

> ### Author Response · Authors · 2023-11-23
> **Response to reviewer ZvAY (continued)**
>
> **Q: “The methodology of training the proposed foundation model is not novel and the authors only apply the stable diffusion model to the satellite images”**
> We want to stress that our work is more than simply “applying stable diffusion to satellite images”. We demonstrate the effectiveness of simply fine-tuning and applying vanilla stable diffusion to satellite images in all of our experiments: in tables 1, 2, 3, 4. All those results demonstrate that StableDiffusion by itself is insufficient to successfully perform generative tasks with satellite data. For single-image generation, SD is unable to capture variation in season, location, or resolution. For super-resolution, it often “hallucinates” results. Similarly, its performance is worse than DiffusionSat for both temporal generation and texas-housing super-resolution.
>
> We clearly demonstrate that applying stable diffusion to satellite images is non-trivial. Given the differences in satellite images with natural data that we mention in the paper (i.e. irregular time intervals, resolutions, multi-spectral bands, correspondence with “global metadata” etc.), we demonstrate the need for a more thoughtful approach towards designing a generative foundation model for satellite images.
>
> DiffusionSat does this; our novelty is in recognizing the important ingredients towards designing the first diffusion-based generative foundation model for remote sensing data. We demonstrate the importance of metadata and associating images with numerical metadata in addition, and even in the absence of, semantic text captions. We design a very general 3D conditioning mechanism that enables DiffusionSat to generate images conditioned on a variety of control signals, including images at different time-steps, resolutions, and multi-spectral bands. Further, we demonstrate, qualitatively and quantitatively, the improvements of our method on a collection of generative tasks. The reviewer’s comments about our methodology “only applying stable diffusion to satellite images” or that it is simply “a combination of StableDiffusion and contolnet” come across as disingenuous.
>
> We hope that our work’s novel approach spurs interest and further activity in an underexplored, but highly important field: solving generative tasks for remote sensing data via foundational models.

---

> ### Author Response · Authors · 2023-11-23
> **References accompanying response to reviewer ZvAY**
>
> [1] Whang, Jay, Mauricio Delbracio, Hossein Talebi, Chitwan Saharia, Alexandros G. Dimakis, and Peyman Milanfar. "Deblurring via stochastic refinement." In Proceedings of the IEEE/CVF Conference on Computer Vision and Pattern Recognition, pp. 16293-16303. 2022.
>
> [2] Luo, Ziwei, Fredrik K. Gustafsson, Zheng Zhao, Jens Sjölund, and Thomas B. Schön. "Refusion: Enabling large-size realistic image restoration with latent-space diffusion models." In Proceedings of the IEEE/CVF Conference on Computer Vision and Pattern Recognition, pp. 1680-1691. 2023.
>
> [3] Kawar, Bahjat, Michael Elad, Stefano Ermon, and Jiaming Song. "Denoising diffusion restoration models." Advances in Neural Information Processing Systems 35 (2022): 23593-23606.
>
> [4] Pinaya, Walter HL, Petru-Daniel Tudosiu, Jessica Dafflon, Pedro F. Da Costa, Virginia Fernandez, Parashkev Nachev, Sebastien Ourselin, and M. Jorge Cardoso. "Brain imaging generation with latent diffusion models." In MICCAI Workshop on Deep Generative Models, pp. 117-126. Cham: Springer Nature Switzerland, 2022.
>
> [5] He, Kaiming, Xinlei Chen, Saining Xie, Yanghao Li, Piotr Dollár, and Ross Girshick. "Masked autoencoders are scalable vision learners." In Proceedings of the IEEE/CVF conference on computer vision and pattern recognition, pp. 16000-16009. 2022.
>
> [6] He, Kaiming, Haoqi Fan, Yuxin Wu, Saining Xie, and Ross Girshick. "Momentum contrast for unsupervised visual representation learning." In Proceedings of the IEEE/CVF conference on computer vision and pattern recognition, pp. 9729-9738. 2020.
>
> [7] Caron, Mathilde, Hugo Touvron, Ishan Misra, Hervé Jégou, Julien Mairal, Piotr Bojanowski, and Armand Joulin. "Emerging properties in self-supervised vision transformers." In Proceedings of the IEEE/CVF international conference on computer vision, pp. 9650-9660. 2021.
>
> [8] Shermeyer, Jacob, and Adam Van Etten. "The effects of super-resolution on object detection performance in satellite imagery." In Proceedings of the IEEE/CVF Conference on Computer Vision and Pattern Recognition Workshops, pp. 0-0. 2019.
>
> [9] Zhang, Chongyuan, Afef Marzougui, and Sindhuja Sankaran. "High-resolution satellite imagery applications in crop phenotyping: an overview." Computers and Electronics in Agriculture 175 (2020): 105584.
>
> [10] Boyle, Sarah A., Christina M. Kennedy, Julio Torres, Karen Colman, Pastor E. Pérez-Estigarribia, and Noé U. de la Sancha. "High-resolution satellite imagery is an important yet underutilized resource in conservation biology." PLoS One 9, no. 1 (2014): e86908.
>
> [11] Nichol, Janet E., Ahmed Shaker, and Man-Sing Wong. "Application of high-resolution stereo satellite images to detailed landslide hazard assessment." Geomorphology 76, no. 1-2 (2006): 68-75.
>
> [12] Li, Xinghua, Zhiwei Li, Ruitao Feng, Shuang Luo, Chi Zhang, Menghui Jiang, and Huanfeng Shen. "Generating high-quality and high-resolution seamless satellite imagery for large-scale urban regions." Remote Sensing 12, no. 1 (2019): 81.
>
> [13] Beck, Anthony, Graham Philip, Maamoun Abdulkarim, and Daniel Donoghue. "Evaluation of Corona and Ikonos high resolution satellite imagery for archaeological prospection in western Syria." antiquity 81, no. 311 (2007): 161-175.
>
> [14] Johansen, Kasper, Nicholas C. Coops, Sarah E. Gergel, and Yulia Stange. "Application of high spatial resolution satellite imagery for riparian and forest ecosystem classification." Remote sensing of Environment 110, no. 1 (2007): 29-44.
>
> [15] Li, Xinghua, Zhiwei Li, Ruitao Feng, Shuang Luo, Chi Zhang, Menghui Jiang, and Huanfeng Shen. "Generating high-quality and high-resolution seamless satellite imagery for large-scale urban regions." Remote Sensing 12, no. 1 (2019): 81.
>
> [16] Xiao, Jieying, Yanjun Shen, Jingfeng Ge, Ryutaro Tateishi, Changyuan Tang, Yanqing Liang, and Zhiying Huang. "Evaluating urban expansion and land use change in Shijiazhuang, China, by using GIS and remote sensing." Landscape and urban planning 75, no. 1-2 (2006): 69-80.
>
> [17] Piyoosh, Atul Kant, and Sanjay Kumar Ghosh. "Semi-automatic mapping of anthropogenic impervious surfaces in an urban/suburban area using Landsat 8 satellite data." GIScience & Remote Sensing 54, no. 4 (2017): 471-494.
>
> [18] Gevaert, Caroline M., Juha Suomalainen, Jing Tang, and Lammert Kooistra. "Generation of spectral–temporal response surfaces by combining multispectral satellite and hyperspectral UAV imagery for precision agriculture applications." IEEE Journal of Selected Topics in Applied Earth Observations and Remote Sensing 8, no. 6 (2015): 3140-3146.
>
> [19] Cha, Keumgang, Junghoon Seo, and Taekyung Lee. "A billion-scale foundation model for remote sensing images." arXiv preprint arXiv:2304.05215 (2023).
>
> [20] Wang, Di, Qiming Zhang, Yufei Xu, Jing Zhang, Bo Du, Dacheng Tao, and Liangpei Zhang. "Advancing plain vision transformer toward remote sensing foundation model." IEEE Transactions on Geoscience and Remote Sensing 61 (2022): 1-15.

---

> > ### Author Response · Authors · 2023-11-23
> > **References, continued**
> >
> > [21] Cong, Yezhen, Samar Khanna, Chenlin Meng, Patrick Liu, Erik Rozi, Yutong He, Marshall Burke, David Lobell, and Stefano Ermon. "Satmae: Pre-training transformers for temporal and multi-spectral satellite imagery." Advances in Neural Information Processing Systems 35 (2022): 197-211.

---

### Official Review · Reviewer_tstN · 2023-11-01

**Soundness:** 2 fair
**Presentation:** 2 fair
**Contribution:** 3 good
**Rating:** 8
**Confidence:** 4

**Summary:**

The paper presents a diffusion based remote sensing model able for the following generative downstream tasks: (i) temporal image generation, (ii) multispectral image super resolution, and (iii) image in-painting.


The work is novel as it represents the first diffusion-based remote sensing model. This is interesting given the multi-spectral nature of remote sensing data and the "image caption" adaptations required for the diffusion backbone. Further, the authors adapted ControlNet architecture to a 3d ControlNet architecture to serve their task.

**Strengths:**

**(S1):** this work presents a novel diffusion-based approach for remote sensing data. It is great to see people extending diffusion to remote sensing data as it is of complex nature given its multi-spectral composition.

**(S2):** this work outlines multiple generative downstream tasks for remote sensing which do go beyond "simple" image generation. This is important because in remote sensing we have no shortage of data and therefore actually not much demand for "simple" image generation.

**Weaknesses:**

**(W1)**: the presented work is very domain specific i.e., remote sensing data. It would be interesting to see if this approach is able to generalize to other datasets of similar multi-spectral data.

**(W2)**: I am confused by the "4. Experiment" section. It is not always straightforward to link the tables and images to the different generative downstream tasks presented. And it seems that some results are missing (i.e., the In-painting section/paragraph is missing entirely)? This point also goes hand in hand with my question Q1, Q2, Q3 below: you mention different downstream tasks in the abstract and the method section, and then the results are not consistently reporting on these downstream tasks (as it seems to me, but maybe I misunderstand something). This is the strongest weakness as it renders the evaluation of the methods to be very difficult for the reader of the submitted paper.

**Questions:**

**(Q1)**: Question for clarification, in your abstract you have mentioned 3 downstream tasks (temporal image generation, multispectral image super resolution, and image in-painting. In your methods section you have mentioned 5 downstream tasks (single-image generation, conditioned on text and metadata, multi-spectral superresolution, temporal prediction, and temporal inpainting). Do you summarize those 5 into the 3 in the abstract? And if yes, why?

**(Q2)**: The "Experiment" section presents results for the following downstream tasks:
"4.1 Single Image Generation", "4.2 Conditional Generation". Then in "4.2 Conditional Generation" results are shown for "super resolution" and "temporal prediction". Finally, the "in-painting" section is missing. This is not in sync with the previously mentioned downstream tasks. Could you clarify the structure of the experiment section?

**(Q3)**: In Table 2, results are presented and compared to other generative methods. Why do you not provide this comparison for Table 1 and Table 3, where results are only compared against non-adapted stable diffusion? Is there any reason for this, if yes, it might be missing in the text body of section 4. I would also recommend to cite the methods you compare to in Table 2.

**(Q4)**: In Fig. 4, which part of the left image is damaged and later in-painted? I see some color-corrections done on the middle image and the ground truth. Is the "damage" done on some specific RGB (or multispectral) channels?

---

> ### Author Response · Authors · 2023-11-20
> **Response to reviewer tstN**
>
> Thank you for your insightful feedback, recognition of DiffusionSat’s novelty and its evaluation on multiple challenging remote-sensing generative tasks. Your suggestions to improve our presentation have been very helpful to make our contributions clearer and our paper more cogent. You may find responses to your questions below:
>
> **Q: Domain specificity of DiffusionSat, exploring its generality beyond remote sensing?**
> Thank you for your suggestion. Our goal is to demonstrate the importance of DiffusionSat as a foundation model tailored to tackle the unique generative tasks provided by remote-sensing datasets. For example, performing super-resolution on widely available multi-spectral Sentinel-2 data can enable object detection and classification for regions of the world which lack high-resolution satellite data [1]. Other studies demonstrate the benefits of high-resolution satellite data for crop-phenotyping [2], for conservation efforts in biology [3], and landslide hazard assessment [4]. Further studies have shown the importance of cloud or shadow removal in satellite images of urban areas [5] towards urban planning.
>
> These references demonstrate the slew of important inverse-problems associated with remote-sensing data which have broad societal implications. They also demonstrate the growing field of using machine-learning techniques for remote-sensing data. However, each of the works above tailors a specific solution to solve the generative task at hand, and therefore that solution often only applies for their use-case. The important advantage and contribution through DiffusionSat is its potential to tackle multiple of these inverse problems at once. We provide researchers and practitioners an easy way to use DiffusionSat to solve a generative task involving remote-sensing data.
>
> Thus, we intentionally evaluate and demonstrate the usefulness of our solution on remote sensing data, with the aim of providing a general-purpose solution and with the hope to spur future interest and research in the field of generative foundation models for this data modality.
>
> We agree that extending and applying DiffusionSat on other forms of multispectral data would be a valuable future line of research inquiry. While we aren’t aware of any such large-scale dataset, we would be happy to accept pointers for the same for future work.
>
> **Q: Clarification on the number/types of downstream tasks**
> Thank you for your observation. We apologise for the structure. We want to clarify that our downstream tasks are: (1) multi-spectral input super-resolution, (2) temporal generation/prediction (3) temporal in-painting for natural disaster datasets of satellite imagery.
>
> The single-image generation tasks (which includes generating single-images given a text prompt and metadata information) are for pre-training DiffusionSat and are not downstream tasks. We first train DiffusionSat on single-image generation, and then use this model for downstream conditional generation on the 3 tasks mentioned above.
>
> **Q: Incomplete in-painting section**
> Thank you for the observation. We apologise; we meant to include a description of this section that was unintentionally omitted in the first submitted version. We have included a description of the task along with the generated samples with a clearer description to aid readability. We describe our changes in the overall response. We hope this adds further clarity.
>
> **Q: Other baselines for superresolution and conditional image generation tasks**
> Thank you for the suggestion. Table 1 is intended to demonstrate the effectiveness of our design choices for pre-training the single-image version of DiffusionSat. We compare with StableDiffusion as the largest, most effective single text-to-image publicly available as of the time of writing (which beats most other generative baselines)[6]. Our results in-turn demonstrate the effectiveness of conditioning on text and metadata for DiffusionSat. The methods in table 2 are task specific and unlike DiffusionSat, they cannot be applied for other relevant generative tasks. For example, STSR or Pix2Pix cannot perform single-image generation given text or metadata.
>
> We have updated Table 3 (now table 2) to include further baselines for super-resolution, based on related work. As can be seen, DiffusionSat outperforms these prior works on the super-resolution tasks. We want to emphasize that the quantitative metrics don’t reveal the full-story- metrics such as SSIM/PSNR/MSE can be high for baselines that produce very fuzzy or unrealistic samples (eg: DBPN [7]). LIPIPS better reflects visual “realism” and our qualitative results also demonstrate the high quality of our generated samples. The drawbacks of SSIM/PSNR are a known problem [8,9].

---

> > ### Author Response · Authors · 2023-11-22
> > **Resolving additional concerns**
> >
> > Dear reviewer,
> > As the discussion deadline approaches, we hope that your concerns are resolved and that you can reconsider our scores. Thank you very much for your time.

---

> ### Author Response · Authors · 2023-11-20
> **Response to reviewer tstN (continued)**
>
> **Q: “why are results are only compared against non-adapted stable diffusion”**
> As mentioned above, we do include additional baselines for comparison. We wanted to emphasize the strength of the StableDiffusion baseline, which has been shown across a variety of natural image generative tasks to outperform other baselines[6], including the ones cited in other tables in our work. Therefore, being able to outperform StableDiffusion by strong margins underscores the benefit of our work: our pre-training on text-metadata specific to satellite imagery unlocks generative models’ potential on the unique properties of satellite images.

---

> ### Author Response · Authors · 2023-11-23
> **References accompanying response to reviewer tstN**
>
> [1] Shermeyer, Jacob, and Adam Van Etten. "The effects of super-resolution on object detection performance in satellite imagery." In Proceedings of the IEEE/CVF Conference on Computer Vision and Pattern Recognition Workshops, pp. 0-0. 2019.
>
> [2] Zhang, Chongyuan, Afef Marzougui, and Sindhuja Sankaran. "High-resolution satellite imagery applications in crop phenotyping: an overview." Computers and Electronics in Agriculture 175 (2020): 105584.
>
> [3] Boyle, Sarah A., Christina M. Kennedy, Julio Torres, Karen Colman, Pastor E. Pérez-Estigarribia, and Noé U. de la Sancha. "High-resolution satellite imagery is an important yet underutilized resource in conservation biology." PLoS One 9, no. 1 (2014): e86908.
>
> [4] Nichol, Janet E., Ahmed Shaker, and Man-Sing Wong. "Application of high-resolution stereo satellite images to detailed landslide hazard assessment." Geomorphology 76, no. 1-2 (2006): 68-75.
>
> [5] Li, Xinghua, Zhiwei Li, Ruitao Feng, Shuang Luo, Chi Zhang, Menghui Jiang, and Huanfeng Shen. "Generating high-quality and high-resolution seamless satellite imagery for large-scale urban regions." Remote Sensing 12, no. 1 (2019): 81.
>
> [6] Rombach, Robin, Andreas Blattmann, Dominik Lorenz, Patrick Esser, and Björn Ommer. "High-resolution image synthesis with latent diffusion models." In Proceedings of the IEEE/CVF conference on computer vision and pattern recognition, pp. 10684-10695. 2022.
>
> [7] Haris, Muhammad, Gregory Shakhnarovich, and Norimichi Ukita. "Deep back-projection networks for super-resolution." In Proceedings of the IEEE conference on computer vision and pattern recognition, pp. 1664-1673. 2018.
>
> [8] Zhang, Richard, Phillip Isola, Alexei A. Efros, Eli Shechtman, and Oliver Wang. "The unreasonable effectiveness of deep features as a perceptual metric." In Proceedings of the IEEE conference on computer vision and pattern recognition, pp. 586-595. 2018.
>
> [9] Saharia, Chitwan, Jonathan Ho, William Chan, Tim Salimans, David J. Fleet, and Mohammad Norouzi. "Image super-resolution via iterative refinement." IEEE Transactions on Pattern Analysis and Machine Intelligence 45, no. 4 (2022): 4713-4726.

---

> ### Comment · Reviewer_tstN · 2023-11-23
> **Feedback to the author's rebuttal**
>
> I'd like to thank the authors for their rebuttal. I think the rebuttal provides clarification and definitely improves my understanding of the paper.
>
> I would like to highlight that I was initially sceptical about the generative downstream tasks focus of the paper, since most of the work that I know in this area focuses on discriminative downstream tasks. However, this work lists indeed relevant generative downstream tasks and might serve as a baseline for future work in this area.
>
> In my opinion the contribution is interesting, novel, and relevant not only to the EO community. I am happy to increase the score and would like to tell the authors to continue working on this interesting topic.

---

> > ### Author Response · Authors · 2023-11-23
> > **Thank you!**
> >
> > Thank you for your consideration and for your vote of confidence in our method! We will definitely press ahead with research in this topic and hope that others are also motivated to do so.

---

### Author Response · Authors · 2023-11-20
**Overall Response to Reviews**

We thank the reviewers for their constructive feedback, comments, and questions. We appreciate the reviewers’ acknowledgement that our work presents a novel diffusion-based generative foundation model (R1, R3), that it has been effectively evaluated across multiple interesting and relevant tasks using remote-sensing data (R1, R3, R4), and that it will make a strong positive impact in the remote sensing community (R3, R4). We also thank the reviewers’ recognition of the impact of providing benchmark datasets to evaluate generative models for remote sensing (R3).

Our work’s main contribution is to provide a large-scale, diffusion-based generative foundational model for remote sensing datasets. The strengths of our method include:
* Incorporating metadata along with semantic textual captions to generate realistic high-resolution satellite images of different object classes, resolutions, locations, and timestamps
* Novel 3D controlnet conditioning which allows DiffusionSat to perform multiple downstream tasks including super-resolution, auto-regressive image sequence generation, and temporal in-painting
* A collection of datasets tailored for generative remote-sensing tasks

The concerns raised by the reviewers include:
* Not enough baselines for some of the conditional generation tasks
* A clear motivation for requiring a generative foundation model for remote-sensing data
* Insufficient generated examples for some of our tasks and datasets
* Unclear presentation/organisation of sections of the paper

To incorporate the reviewers’ feedback, we have included the following in our revision:

**Baselines for super-resolution and temporal generation**
We include quantitative results from DBPN for our super-resolution task. DBPN [1] represents a prior state-of-the-art superresolution method for natural images. Our methods demonstrate improvements over these baselines on the difficult task of fMoW-Sentinel (10m-60m GSD) → fMoW RGB super-resolution (0.5m-1m GSD).

For temporal prediction, we compare against Masked Conditional Video Diffusion (MCVD) [2], a state-of-the-art method for conditional video generation, and training our 3D controlnet on StableDiffusion 2.1 weights. We find through both quantitative and qualitative results that DiffusionSat outperforms these methods for past and future prediction.

**Qualitative comparisons via generated samples**
We have included generated samples for our conditional generation tasks.

For super-resolution, our qualitative results demonstrate greater accuracy and sample quality even when small details in the conditioning multi-spectral input are difficult to view. StableDiffusion’s super-resolution suffers from hallucination, and other baselines don’t produce sharp or realistic details.

For temporal prediction,  a key strength of DiffusionSat is its ability to understand the correlation between metadata and images, which results in generated outputs that reflect season, urban development, and resolution (GSD). Baselines such as STSR and MCVD frequently conflate seasons, eg: producing outputs that mix summer-time grass and winter-time snow, or simply reproduce an image from the input conditioning sequence. DiffusionSat attempts to understand trends in housing/road development and even crop-type maturity in its predictions, which are closer to ground truth than the baselines. We also include _unconditional_ temporal generation results in the appendix.

We also show DiffusionSat’s capability to in-paint damaged regions for a variety of disasters. Flooded roads, the rubble from houses, and terrain is identified and reconstructed to its pre-disaster state. DiffusionSat can also predict the influence of types of natural disasters on a given satellite image. We hope that this tool is useful to researchers and first-responders who wish to understand damaged or obscured structures in a region in the event of a disaster (eg: which roads are flooded/blocked, which houses were razed, how the landscape changes due to disaster etc.)

Some other changes include:
* Updated DiffusionSat results. We find that training our 3D ControlNets for longer (20k -> 40k iterations) boosts performance for temporal generation. These results have been updated.
* Clearer motivation for DiffusionSat’s strengths as a generative foundation model and updated writing/organisation for sections of the paper to increase clarity.
* A preliminary investigation of geographical bias in the appendix.

Bottlenecked by academic computational resources and expensive experiments (eg: single-image DiffusionSat pre-training takes 5 days on 8x A100 NVIDIA GPUs), we have focused on including the most salient additions to demonstrate the effectiveness of DiffusionSat- we hope the reviewers understand.

The following comments include an in-depth response to each reviewer’s specific concerns and comments.

Link to the revised paper: https://openreview.net/pdf?id=I5webNFDgQ

---

> ### Author Response · Authors · 2023-11-20
> **References accompanying response**
>
> Reference:
> [1] Haris, Muhammad, Gregory Shakhnarovich, and Norimichi Ukita. "Deep back-projection networks for super-resolution." In Proceedings of the IEEE conference on computer vision and pattern recognition, pp. 1664-1673. 2018.
>
> [2] Voleti, Vikram, Alexia Jolicoeur-Martineau, and Chris Pal. "MCVD-masked conditional video diffusion for prediction, generation, and interpolation." Advances in Neural Information Processing Systems 35 (2022): 23371-23385.

---

### Meta-Review · Area_Chair_iWNw · 2023-12-06

**Metareview:**

This paper applies diffusion models to remote sensing. It trains a large generative model on high-resolution remote sensing data. It considers both unconditional and conditional cases.

This is a pure application paper and reviewers did not reach an agreement after discussion. In particular, Reviewer ZvAY thought that it only combines the stable diffusion and controllnet as well as some new conditional inputs and kept the score as "reject".

I think this this a borderline paper. I am recommending an acceptance but I wouldn't mind if the paper gets rejected.

**Justification For Why Not Higher Score:**

Reviewer ZvAY thought that it only combines the stable diffusion and controllnet as well as some new conditional inputs and kept the score as "reject".

**Justification For Why Not Lower Score:**

I think this this a borderline paper. I am recommending an acceptance but I wouldn't mind if the paper gets rejected.

---

### Decision · Program_Chairs · 2024-01-16

Accept (poster)